# SQALER: Scaling Question Answering by Decoupling Multi-Hop and Logical Reasoning

**Mattia Atzeni**
IBM Research, EPFL
Switzerland
atz@zurich.ibm.com

**Jasmina Bogojeska**
IBM Research
Switzerland
jbo@zurich.ibm.com

**Andreas Loukas**
EPFL
Switzerland
andreas.loukas@epfl.ch

## Abstract

State-of-the-art approaches to reasoning and question answering over knowledge graphs (KGs) usually scale with the number of edges and can only be applied effectively on small instance-dependent subgraphs. In this paper, we address this issue by showing that multi-hop and more complex logical reasoning can be accomplished separately without losing expressive power. Motivated by this insight, we propose an approach to multi-hop reasoning that scales linearly with the number of relation types in the graph, which is usually significantly smaller than the number of edges or nodes. This produces a set of candidate solutions that can be provably refined to recover the solution to the original problem. Our experiments on knowledge-based question answering show that our approach solves the multi-hop MetaQA dataset, achieves a new state-of-the-art on the more challenging WebQuestionsSP, is orders of magnitude more scalable than competitive approaches, and can achieve compositional generalization out of the training distribution.

## 1 Introduction

Reasoning, namely the ability to infer conclusions and draw predictions based on existing knowledge, is a hallmark of human intelligence. Infusing the same ability into machine learning models has been a major challenge [34, 29] and has historically required complex systems made of several hand-crafted or learned components [47, 19]. Recently, the paradigm has shifted to deep learning approaches [43, 42], where neural networks are used to reason over structured knowledge or a text corpus. In this work, we assume that the source of knowledge is a structured knowledge graph (KG) and we tackle the problem of *knowledge-based question answering* (KBQA), namely finding answers to natural language queries involving multi-hop and logical reasoning over the KG.

Answering queries over a knowledge graph involves many challenges, among which scalability is a major issue. Real-world KGs often contain millions of nodes and even a 2-hop neighborhood of the entities mentioned in the query may comprise tens of thousands of nodes. Many state-of-the-art approaches [41, 42, 39] address the challenge of scalability by building small query-dependent subgraphs. To this end, they usually use simple heuristics [41] or, in some cases, iterative procedures based on learned classifiers [42]. This preprocessing step is usually needed because each forward pass in end-to-end neural networks for KBQA scales at least linearly with the number of edges in the subgraph. Training neural networks involves repeated evaluation, which renders even a linear complexity impractical for graphs of more than a few tens thousands of nodes.

In order to address this issue, we introduce a novel approach called SQALER (Scaling Question Answering by Leveraging Edge Relations). The method first learns a model that generates a set of candidate answers (entities in the KG) by *multi-hop reasoning*: the candidate solutions are obtained by starting from the set of entities mentioned in the question and seeking those that provide an answer by chained relational following operations. We refer to this module as the *relation-level* model.

We show that this multi-hop reasoning step can be done efficiently and provably generates a set of candidates including all the actual answers to the original question. **SQALER** then uses a second-stage *edge-level* model that recovers the real answers by performing logical reasoning on a subgraph in the vicinity of the candidate solutions. A visual summary of our approach is depicted in Figure 1.

The main contributions and takeaway messages of this work are the following:

1. KBQA can be addressed by first performing multi-hop reasoning on the KG and then refining the result with more sophisticated logical reasoning without losing expressive power (we will elaborate this claim in more details in Section 2.3).

2. Multi-hop reasoning can be accomplished efficiently with a method that scales linearly with the number of relation types in the KG, which are usually significantly fewer than the number of facts or entities.

In the remainder of the paper, we first provide an extensive overview of our approach and a theoretical analysis of the expressive power and the computational complexity of **SQALER**. Our experimental results show that **SQALER** achieves better reasoning performance than state-of-the-art approaches, generalizes compositionally out of the training distribution, and scales to the size of real-world knowledge graphs with millions of entities.

## 2   Scaling KBQA with relation and edge-level reasoning

This section provides a detailed description of our approach. We start by defining the problem formally and giving an intuitive overview of **SQALER**. Then, we discuss the approach in more details and we analyze its computational complexity and expressive power

**Problem statement.**   We denote a knowledge graph as $\mathcal{G} = (\mathcal{V}, \mathcal{R}, \mathcal{E})$, where $v \in \mathcal{V}$ represents an entity or node in $\mathcal{G}$, $r \in \mathcal{R}$ is a relation type, and we write $v \xrightarrow{r} v'$ to denote an edge in $\mathcal{E}$ labeled with relation type $r \in \mathcal{R}$ between two entities $v, v' \in \mathcal{V}$. We extend the same notation to sets of nodes by writing $\mathcal{V}_i \xrightarrow{r} \mathcal{V}_j$ if $\mathcal{V}_j = \{v_j \in \mathcal{V} \mid v_i \xrightarrow{r} v_j, v_i \in \mathcal{V}_i\}$. Given a knowledge graph $\mathcal{G} = (\mathcal{V}, \mathcal{R}, \mathcal{E})$ and a natural language question $Q$, expressed as a sequence of tokens $Q = (q_1, q_2, \ldots, q_{|Q|})$, in *knowledge-based question answering* the objective is to identify a set of nodes $\mathcal{A}_Q \subseteq \mathcal{V}$ representing the correct answers to $Q$. Following previous work [41, 42, 43], we assume that the set of entities mentioned in the question $\mathcal{V}_Q \subseteq \mathcal{V}$ is given. These nodes are also called the *anchor nodes* of the question and in practice are commonly obtained using an entity-linking module.

**Overview.**   KBQA can be cast as an entity seeking problem on $\mathcal{G}$ by translating $Q$ into a set of nodes $\mathcal{V}_Q \subseteq \mathcal{V}$ (the starting points of the search) and seeking for nodes that provide an answer [41, 42, 43]. Attempting to find $\mathcal{A}_Q$ directly on $\mathcal{G}$ is prohibitive in practice, as even the most efficient graph-based neural networks generally scale at least linearly with the number of edges. Our approach mitigates this issue by breaking the problem in two subproblems.

(a) We first utilize a *relation-level* model $\phi$ to obtain a set of *candidate answers* $\tilde{\mathcal{A}}_Q$, such that $\mathcal{A}_Q \subseteq \tilde{\mathcal{A}}_Q$. We refer to $\phi$ as "*relation-level*" because, as we will see, it operates on the *coalesced graph*, a simplified representation of $\mathcal{G}$, where edges of the same relation type are coalesced. The coalesced graph is constructed before training and incurs a one-time linear cost. By exploiting it during training, the relation-level model scales with the number of (distinct) relation types in the KG, which are usually significantly fewer than the number of edges or entities.

(b) The candidate answers are then refined using an *edge-level* model $\psi$ applied on a subgraph $\mathcal{G}(\tilde{\mathcal{A}}_Q)$ of the original knowledge graph in the vicinity of $\tilde{\mathcal{A}}_Q$. We should note that the refining step is not always necessary. Indeed, we found that a relation-level model is sufficient to perfectly solve tasks like multi-hop question answering [50]. Figure 1 shows an overview of our approach.

### 2.1   Relational coalescing for efficient knowledge seeking

Our approach relies on a relation-level model $\phi$ that operates as a knowledge seeker in $\mathcal{G}$. The model identifies a node $v$ as a candidate $v \in \tilde{\mathcal{A}}_Q$ based on the sequence of relations that connect it with $\mathcal{V}_Q$.

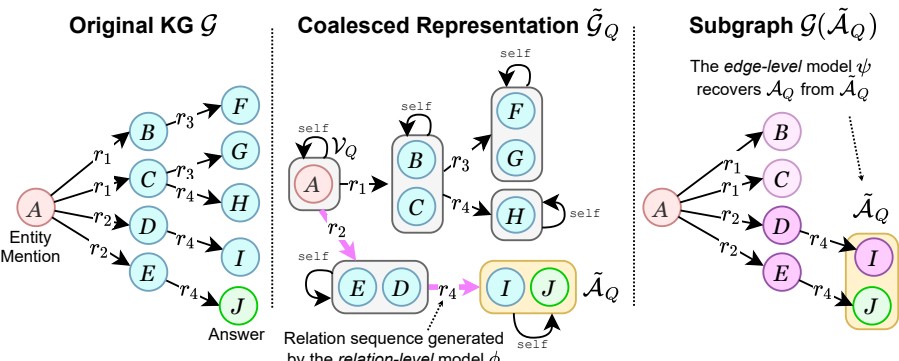

Figure 1: Overview of our approach. A *relation-level* model operates on a coalesced representation of the original KG to generate a set of candidate answers $\tilde{\mathcal{A}}_Q$. This approximate solution is then refined by an *edge-level* model applied on a subgraph of the original KG.

This can be achieved by using a neural network $\phi$ to predict how likely it is that the correct answer is reached from $\mathcal{V}_Q$ by following a sequence of relations $R$.

**Reachability.** To define how our method works, it will help to formalize the concept of reachability. Let $R = (r_1, \ldots, r_{|R|})$ be a sequence of relations. We say that "$v$ is $R$-*reachable from* $\mathcal{V}_Q$" if there exists a path $P = (v_1, \ldots, v_{|R|}, v)$ in $\mathcal{G}$ such that:

$$v_1 \in \mathcal{V}_Q \quad \text{and} \quad v_i \xrightarrow{r_i} v_{i+1} \quad \text{for every} \quad i = 1, \ldots, |R|.$$

That is, we can reach $v$ by starting from a node in $\mathcal{V}_Q$ and following a sequence of edges with relation types $R$. We also denote by $reach_{\mathcal{G}}(\mathcal{V}_Q, R)$ the set of nodes that are $R$-reachable from $\mathcal{V}_Q$:

$$reach_{\mathcal{G}}(\mathcal{V}_Q, R) = \{v \in \mathcal{V} \mid v \text{ is } R\text{-reachable from } \mathcal{V}_Q\}.$$

**Relational coalescing.** Given a knowledge graph $\mathcal{G}$, a question $Q$, and a set of entity mentions $\mathcal{V}_Q$, we consider a representation of the graph $\tilde{\mathcal{G}}_Q = (\tilde{\mathcal{V}}_Q, \tilde{\mathcal{R}}_Q, \tilde{\mathcal{E}}_Q)$, which allows us to efficiently compute sets of nodes that are reachable from $\mathcal{V}_Q$. We refer to this representation as the question-dependent coalesced KG, because edges with the same relation type are coalesced, as shown in Figure 1. The nodes of $\tilde{\mathcal{G}}_Q$ are sets of nodes of $\mathcal{G}$ that are reachable from $\mathcal{V}_Q$ by following any possible sequence of relations originating from $\mathcal{V}_Q$. The graph $\tilde{\mathcal{G}}_Q$ has an edge $\mathcal{V}_i \xrightarrow{r} \mathcal{V}_j$ if $\mathcal{V}_j$ is the set of nodes that are reachable from $\mathcal{V}_i$ by following relation $r$. For convenience, we include a relation type $\texttt{self} \in \tilde{\mathcal{R}}_Q$ to denote self loops. We refer the reader to Appendix A for a formal definition of $\tilde{\mathcal{G}}_Q$. The coalesced graph can be precomputed once as a preprocessing step for each question $Q$ and incurs a one-time linear cost. In practice, however, we do not need to compute and store all the nodes in $\tilde{\mathcal{G}}_Q$ but only edge labels. This makes learning efficient because each forward/backward pass scales with the number of relation types and does not depend on the number of nodes or edges in the KG.

**Knowledge seeking in $\tilde{\mathcal{G}}_Q$.** The coalesced graph allows us to provide approximate answers to input questions in an efficient manner. Specifically, we seek $k \geq 1$ sequences of relations $R_i^\star$, such that:

$$\mathcal{A}_Q \subseteq \tilde{\mathcal{A}}_Q = \bigcup_{i=1}^{k} reach_{\mathcal{G}}(\mathcal{V}_Q, R_i^\star).$$

We can achieve this by using a model $\phi$ that only considers relation sequences originating from $\mathcal{V}_Q$. The model predicts the likelihood $\phi : \tilde{\mathcal{E}}_Q \to [0, 1]$ of following a certain edge in a relation sequence from $\mathcal{V}_Q$ to $\tilde{\mathcal{A}}_Q$. Then, given $R = (r_1, \ldots, r_{|R|})$ and a node in the coalesced graph $\mathcal{V}_Q$, we can compute the likelihood of $R$ by multiplying the likelihood of all edges traversed by $R$ in $\tilde{\mathcal{G}}_Q$:

$$\mathsf{P}(R \mid Q, \tilde{\mathcal{G}}_Q, \mathcal{V}_Q) \propto \prod_{i=1}^{|R|} \phi(reach_{\mathcal{G}}(\mathcal{V}_Q, R_{1 \to i-1}), r_i, reach_{\mathcal{G}}(\mathcal{V}_Q, R_{1 \to i}) \mid Q),$$

where $R_{1 \to i} = (r_1, \ldots, r_i)$ is the subsequence of $R$ up to the $i$-th relation. We generate $\tilde{\mathcal{A}}_Q$ by selecting the top $k$ relation sequences $R_i^\star$ with maximum likelihood $\mathsf{P}(R_i^\star \mid Q, \tilde{\mathcal{G}}_Q, \mathcal{V}_Q)$. This can be done by an efficient search algorithm, such as beam search starting from $\mathcal{V}_Q$. Then, we compute $\tilde{\mathcal{A}}_Q$ as the union of all target nodes of the selected relation sequences. More details about the knowledge-seeking algorithm are provided in Appendix $B$.

## 2.2 Refining the solution on the original KG

In certain cases, like multi-hop question answering [50], the set of candidate answers $\tilde{\mathcal{A}}_Q$ may already be a reasonable estimate of $\mathcal{A}_Q$. We will substantiate this claim experimentally in Section 4. In general, however, we recover $\mathcal{A}_Q$ by using an *edge-level* model $\psi$ applied on a subgraph $\mathcal{G}(\tilde{\mathcal{A}}_Q)$ of $\mathcal{G}$. Specifically, we construct $\mathcal{G}(\tilde{\mathcal{A}}_Q)$ as the subgraph induced by the set of nodes $\mathcal{V}(\tilde{\mathcal{A}}_Q)$, which includes all nodes visited when following the top-$k$ relation sequences along with their neighbors (see Figure 1 for an example). Any existing method for KBQA can be used to instantiate $\psi$ by running it on $\mathcal{G}(\tilde{\mathcal{A}}_Q)$ rather than $\mathcal{G}$. We opted to use a Graph Convolutional Network (GCN) conditioned on the input question with the same architecture as in [41]. The edge-level model is constrained to predict an answer among the candidates generated by the relation-level model.

## 2.3 Analysis of scalability and expressive power

This section provides a scalability analysis of our approach and shows that the relation-level model scales linearly with the number of relation types in the graph. Then, we analyse the expressive power of **SQALER** and we show the class of supported logical queries.

**Computational complexity.** As mentioned, we do not evaluate the likelihood $\phi$ for all edges in $\tilde{\mathcal{G}}_Q$, but we generate the most likely relation sequences using a knowledge-seeking procedure based on the beam search algorithm. At any given time step, only the $\beta$ most likely relation sequences are retained and further explored at the next iteration. Hence, the time complexity required by our algorithm is $\mathcal{O}(\tau_{max} \cdot \beta \cdot d^+_{max}(\tilde{\mathcal{G}}_Q))$, where $\tau_{max}$ is the maximum allowed number of decoding time steps and $d^+_{max}(\tilde{\mathcal{G}}_Q)$ is the maximum outdegree of $\tilde{\mathcal{G}}_Q$. Note that $d^+_{max}(\tilde{\mathcal{G}}_Q)$ is bounded by the number of relations in the graph, whereas $\tau_{max}$ and $\beta$ are constant parameters of the algorithm and are usually small. This gives a time complexity of:

$$\mathcal{O}(\tau_{max} \cdot \beta \cdot |\mathcal{R}|) = \mathcal{O}(|\mathcal{R}|).$$

Hence, the knowledge-seeking algorithm scales linearly with the number of relations in the KG. The space complexity is also $\mathcal{O}(\tau_{max} \cdot \beta \cdot |\mathcal{R}|)$. A more detailed analysis is provided in Appendix B.

**Expressive power.** Given a natural language question $Q$, we can represent the inferential chain needed to obtain $\mathcal{A}_Q$ from $\mathcal{V}_Q$ as a logical query $\mathsf{Q}$ on $\mathcal{G}$. As an example, the question in Figure 2, *"Who starred in films directed by George Lucas?"*, can be represented by the logical query: $\mathsf{Q}[\mathsf{V}_?] = \mathsf{V}_?.\exists\mathsf{V} : \mathsf{Directed}(\mathsf{George\_Lucas}, \mathsf{V}) \land \mathsf{Starred}(\mathsf{V}, \mathsf{V}_?)$. We denote with $\mathsf{V}_?$ the target variable of the query and we say that $v \in \mathcal{V}$ satisfies $\mathsf{Q}$ if $\mathsf{Q}[v] = \mathsf{True}$. A query $\mathsf{Q}$ is an *existential positive first-order (EPFO) query* if it involves the existential quantification ($\exists$), conjunction ($\land$), and disjunction ($\lor$) [12] of literals corresponding to relations in the KG. Each literal is of the form $r(\mathsf{V}, \mathsf{V}')$, where $\mathsf{V}$ is either a node in $\mathcal{V}_Q$ or an existentially quantified bound variable, and $\mathsf{V}'$ is either an existentially quantified bound variable or the target variable. A literal $r(\mathsf{V}, \mathsf{V}')$ is satisfied if $\mathsf{V} \xrightarrow{r} \mathsf{V}'$, for $r \in \mathcal{R}$. Any EPFO query can be represented in *disjunctive normal form* (DNF) [15], namely as a disjunction of conjunctions. Note that, we do not consider queries with universal quantification ($\forall$), as we assume that in real-world KGs no entity connects to all the others. Then, the following proposition holds for any knowledge graph and EPFO query.

**Proposition 1.** *Let $\mathcal{G} = (\mathcal{V}, \mathcal{R}, \mathcal{E})$ be a knowledge graph and $\mathcal{V}_Q \subseteq \mathcal{V}$ denote a set of entities in $\mathcal{G}$. Let $\mathsf{Q}$ be a valid existential positive first-order query on $\mathcal{G}$ and let $n_\lor$ be the number of disjunction operators in the disjunctive normal form of $\mathsf{Q}$. Then, there exist $k \leq n_\lor + 1$ sequences of relations $R_i^\star \in \mathcal{R}^*$ such that:*

$$\mathcal{A}_Q \subseteq \bigcup_{i=1}^{k} reach_{\mathcal{G}}(\mathcal{V}_Q, R_i^\star),$$

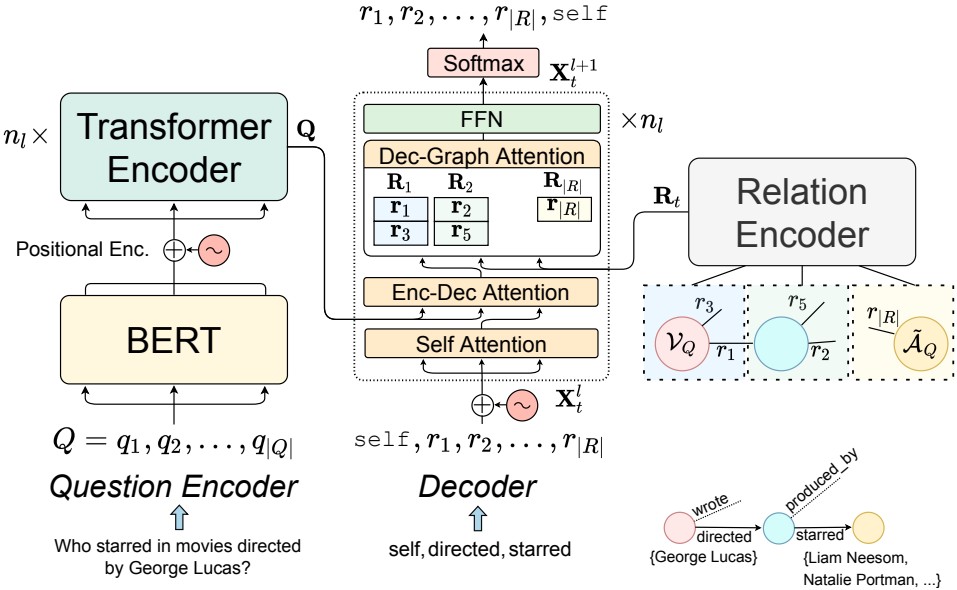

Figure 2: Architecture of the **SQALER** relation-level model. A question encoder is used to obtain a representation of tokens in the input natural language question. Then a graph-guided decoder is applied to obtain the likelihood of output relation sequences. The decoder is constrained to only attend to valid relations according to the structure of the coalesced KG.

*where $\mathcal{A}_Q = \{v \in \mathcal{V} \mid \mathsf{Q}[v] = \mathsf{True}\}$ is the denotation set of $\mathsf{Q}$, namely the entities satisfying $\mathsf{Q}$.*

This shows that sampling $n_\vee + 1$ sequences of relations allows generating a set of candidate answers $\tilde{\mathcal{A}}_Q$ that does not miss any of the real answers $\mathcal{A}_Q$. Then, assuming that the edge-level model $\psi$ can recover $\mathcal{A}_Q$ from $\tilde{\mathcal{A}}_Q$, our approach can be used to answer any EPFO query on $\mathcal{G}$. More details about the expressive power of **SQALER** and the proof of Proposition 1 are provided in Appendix C.

## 3 Architecture of the relation-level model

For the relation-level model $\phi$, we propose an auto-encoder, where the the decoder is constrained to follow sequences of relations in the coalesced representation $\tilde{\mathcal{G}}_Q$. We train the network with weak supervision, assuming that a sequence of relations is correct if it reaches a set of candidate answers $\tilde{\mathcal{A}}_Q$ that is the smallest reachable superset of the $\mathcal{A}_Q$. We found it useful to *pretrain* the model in order to infuse knowledge from the KG. In this case, we train the model to predict a path in the KG, given the representations of the source and target nodes. More details about training strategies are given in Appendix D.

The architecture of the model (see Figure 2) includes three main components: a *question encoder*, a *relation encoder* and a *graph-guided decoder*. We explain each one below.

**Question encoder.** The encoder receives as input a natural language question, which comprises a sequence of tokens $Q = (q_1, q_2, \ldots, q_{|Q|})$. The question is encoded using a pre-trained BERT [18] model and processed with the same positional encoding technique used in [44]. The resulting embeddings are then fed into $n_l = 3$ transformer encoder layers [44]. This results in a matrix $\mathbf{Q} \in \mathbb{R}^{|Q|+1 \times d_{model}}$, where the first row vector is an overall representation of the whole query $Q$ (derived from the embedding of the [CLS] token introduced by BERT) and each remaining row represents the final $d_{model}$-dimensional encoding of a token in the input question.

**Relation encoder.** The relation encoder produces a representation $\mathbf{r} \in \mathbb{R}^{d_{model}}$ for each relation type $r \in \mathcal{R}$. We decided to encode relations based on their surface form, with the same pre-trained BERT model used in the question encoder. In this case, only the embedding of the [CLS] token is used in order to get the final representation $\mathbf{r}$ of each relation type $r \in \mathcal{R}$. At inference time, or in

case the BERT model is not fine-tuned, the embeddings of the relations can be precomputed as a preprocessing step to improve the efficiency of the approach.

**Graph-guided decoder.** The decoder's job is to predict a sequence of relations leading from $\mathcal{V}_Q$ to $\tilde{\mathcal{A}}_Q$ in $\tilde{\mathcal{G}}_Q$. At any time step $t$, it receives as input a sequence of relations $R_t = (\texttt{self}, r_1, \ldots, r_{t-1})$ and predicts the next relation $r_t$ (self is used as a special token to denote the start of decoding). Note that the input sequence uniquely determines a node $\mathcal{V}_t$ in the graph $\tilde{\mathcal{G}}_Q$, namely the node reachable from $\mathcal{V}_Q$ by following $R_t$. The decoder thus selects $r_t$ by choosing amongst the outgoing edges $\tilde{\mathcal{E}}_t$ of $\mathcal{V}_t$. We use the same number of layers $n_l$ both for the question encoder and the decoder. Let $\mathbf{X}_t^l = [\mathbf{x}_0^l, \ldots, \mathbf{x}_{t-1}^l]^\top \in \mathbb{R}^{t \times d_{model}}$ denote the hidden state of the $l$-th layer of the decoder preceding time step $t$. Note that $\mathbf{X}_t^0$ is the representation of the sequence $R_t$, obtained by using the relation encoder described above and the same positional encoding technique used in the question encoder. For each decoder layer, we perform self-attention over the target sequence $\mathbf{X}_t^l$ by computing:

$$\bar{\mathbf{x}}_t^l = Attention(\mathbf{x}_t^l, \mathbf{X}_t^l, \mathbf{X}_t^l),$$

where *Attention* is a function that performs multi-head scaled dot-product attention [44] with skip connections and layer normalization [4]. The above step allows each relation in the decoded sequence to attend to all the others predicted up to time step $t$. We then let the result attend to the question as:

$$\bar{\mathbf{x}}_t^{Q,l} = Attention(\bar{\mathbf{x}}_t^l, \mathbf{Q}, \mathbf{Q}).$$

This is done in order to update the current state of the decoder based on the input question. Next, let $\mathbf{R}_t \in \mathbb{R}^{|\tilde{\mathcal{E}}_t| \times d_{model}}$ denote the encoding of the relations labeling all edges in $\tilde{\mathcal{E}}_t$. We constrain the decoded sequence to follow the structure of the graph by attending only to valid relations as follows:

$$\bar{\mathbf{x}}_t^{R,l} = Attention(\bar{\mathbf{x}}_t^{Q,l}, \mathbf{R}_t, \mathbf{R}_t).$$

We get the hidden state of the next layer $\mathbf{x}_t^{l+1}$ by processing the result with a feed forward network. The model outputs a categorical distribution $\phi(e \mid Q) \in [0, 1]$ over the edges $e \in \tilde{\mathcal{E}}_t$, by applying a softmax function as follows:

$$\phi(\mathcal{V}_i \xrightarrow{r} \mathcal{V}_j \mid Q) = \frac{\exp(\mathbf{r}^\top \mathbf{x}_t^{n_l})}{\sum_{\mathcal{V}_i' \xrightarrow{r'} \mathcal{V}_j' \in \tilde{\mathcal{E}}_t} \exp(\mathbf{r'}^\top \mathbf{x}_t^{n_l})},$$

where $\mathbf{x}_t^{n_l}$ is the output of the final layer of the decoder, whereas $\mathbf{r}$ and $\mathbf{r}'$ denote the representations of relations $r$ and $r'$ respectively.

## 4 Experiments

This section presents an evaluation of our approach with respect to both reasoning performance and scalability. We first show that SQALER reaches state-of-the-art results on popular KBQA benchmarks and can generalize compositionally out of the training distribution. Then, we demonstrate the scalability of our approach on KGs with millions of nodes. We refer the reader to Appendix E for more details about the experiments.

### 4.1 Experimental setup

**Datasets.** We evaluate the reasoning performance of our approach on *MetaQA* [50] and *WebQuestionsSP* [49]. *MetaQA* includes multi-hop questions over the WikiMovies KB [35] and we consider both 2-hop (**MetaQA 2**) and 3-hop (**MetaQA 3**) queries. *WebQuestionsSP* (**WebQSP**) comprises more complex questions answerable over a subset of Freebase [21, 7], a large KG with millions of entities. We further assess the compositional generalization ability of SQALER on the *Compositional Freebase Questions* (*CFQ*) dataset [28]. Each question in *CFQ* is obtained by composing primitive elements (*atoms*). Whereas the training and test distribution of atoms are similar, the test set contains different *compounds*, namely new ways of composing these atoms. *CFQ* comprises three dataset splits (**MCD1**, **MCD2**, and **MCD3**), with maximal compound divergence (MCD) between the training and test distributions. We refer the reader to Appendix E.1 for an extensive description of the datasets.

**Evaluation protocol.** In our experiments on *MetaQA* and *WebQuestionsSP*, we assess the performance of three variants of our approach: (a) a version that only makes use of the relation-level model without the refinement step (**SQALER – Unrefined**), (b) a model that utilizes a key-value memory network to identify the correct answers from the candidates (**SQALER – KV-MemNN**), and (c) a model that uses a GNN architecture for the refinement step (**SQALER – GNN**), as explained in Section 2.2. Following previous work [41, 42, 43, 39], we evaluate the models based on the *Hits@1* metric. On the *CFQ* dataset, we evaluate the accuracy of the refined model with the GNN based on whether it predicts exactly the same answers given by the corresponding SPARQL query.

## 4.2 Main results

**KBQA Performance.** Table 1 summarizes the results of our experiments on the two benchmark datasets. For the two multi-hop *MetaQA* datasets, we achieve state-of-the-art performance by only using the relation-level model of **SQALER**. As shown in Table 1, **SQALER** outperforms all the baselines on **MetaQA 3**, demonstrating the ability of our approach to perform multi-hop reasoning over a KG. For the more complex questions in the *WebQuestionsSP* dataset, the unrefined **SQALER** model achieves better performance than all but one (**EmQL**) of the baselines. To achieve such performance, however, EmQL creates a custom set of logical operations tailored towards the specifics of the target KG and the kind of questions in the dataset, while our approach is agnostic with respect to such details. Combining the relation and edge-level models improves the performance on **WebQSP**. In particular, **SQALER – GNN** outperforms all considered baselines on the three datasets.

Table 1: Hits@1 on *MetaQA* and *WebQuestionsSP*

|  | MetaQA 2 | MetaQA 3 | WebQSP |
|---|---|---|---|
| **KV-MemNN** [35] | 82.7 | 48.9 | 46.7 |
| **GRAFT-Net** [41] | 94.8 | 77.7 | 70.3 |
| **ReifKB + mask** [10] | 95.4 | 79.7 | 52.7 |
| **PullNet** [42] | 99.9 | 91.4 | 69.7 |
| **EmbedKGQA** [39] | 98.8 | 94.8 | 66.6 |
| **EmQL** [43] | 98.6 | 99.1 | 75.5 |
| **SQALER – Unrefined** | 99.9 | 99.9 | 70.6 |
| **SQALER – KV-MemNN** | 99.9 | 99.9 | 72.1 |
| **SQALER – GNN** | 99.9 | 99.9 | 76.1 |

**Compositional generalization.** In order to evaluate the compositional generalization ability of **SQALER**, we performed additional experiments on the *CFQ* dataset. Table 2 shows the accuracy on the three MCD splits and the mean accuracy (**MCD-mean**) in comparison to the other methods in the leaderboard. Note that the other approaches address a semantic parsing task and require additional supervision, as they are trained to predict the target query. On the other hand, we aim to predict directly the set of answers to the input question. The experiment shows that **SQALER** is able to achieve compositional generalization with an accuracy comparable to the state-of-the-art model on *CFQ* for semantic parsing.

**Subgraph extraction.** We analyzed the candidate solutions produced by the relation-level model in order to evaluate the suitability of our approach to building small question subgraphs that are likely to contain the answers to a natural language question. For this purpose, we computed the precision and recall of the set of candidate answers with varying number of relation sequences sampled by the relation-level model. Figure 3 shows the top relation sequences predicted by the relation-level model on two questions from the test set of *WebQuestionsSP*. The precision and recall curves are shown in Figure 4. As expected, on *MetaQA* the recall is high for all values of $k$, because selecting the most likely sequence of relations is sufficient to solve the multi-hop question answering task. On *WebQuestionsSP*, only 3 sequences of relations are sufficient to obtain a recall of $0.91$, and we can improve it to $0.95$ by generating still small subgraphs consisting of only 10 sequences of relations.

Table 2: Accuracy and 95% confidence interval on the *CFQ* dataset

| | MCD1 | MCD2 | MCD3 | MCD-mean |
|---|---|---|---|---|
| **LSTM + Attention** [28, 27, 5] | $0.289 \pm 0.018$ | $0.050 \pm 0.008$ | $0.108 \pm 0.006$ | $0.149 \pm 0.011$ |
| **Transformer** [28, 44] | $0.349 \pm 0.011$ | $0.082 \pm 0.003$ | $0.106 \pm 0.011$ | $0.179 \pm 0.009$ |
| **Universal Transformer** [28, 17] | $0.374 \pm 0.022$ | $0.081 \pm 0.016$ | $0.113 \pm 0.003$ | $0.189 \pm 0.014$ |
| **Evolved Transformer** [20, 40] | $0.424 \pm 0.010$ | $0.093 \pm 0.008$ | $0.108 \pm 0.002$ | $0.208 \pm 0.007$ |
| **T5-11B** [36, 20] | $0.614 \pm 0.048$ | $0.301 \pm 0.022$ | $0.312 \pm 0.057$ | $0.409 \pm 0.043$ |
| **T5-11B-mod** [20, 22] | $0.616 \pm 0.124$ | $0.313 \pm 0.128$ | $0.333 \pm 0.023$ | $0.421 \pm 0.091$ |
| **HPD** [23] | $0.720 \pm 0.075$ | $0.661 \pm 0.064$ | $0.639 \pm 0.057$ | $0.673 \pm 0.041$ |
| **SQALER − GNN** | $0.734 \pm 0.039$ | $0.653 \pm 0.040$ | $0.627 \pm 0.045$ | $0.671 \pm 0.041$ |

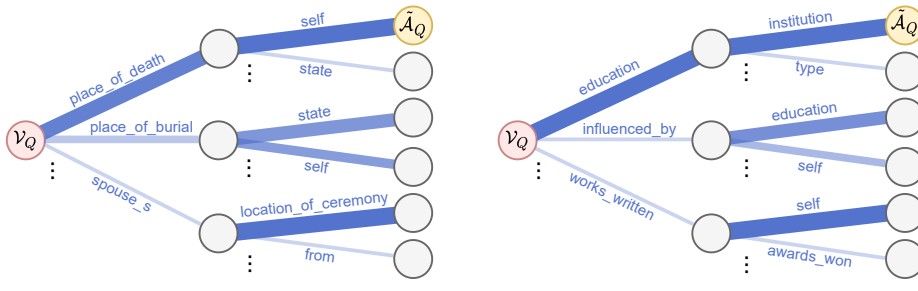

Figure 3: Attention weights given by the relation-level model to the edges of the coalesced graph for two questions in *WebQuestionsSP*. Thicker and darker edges represent higher attention weights.

## 4.3 Efficiency and scalability

We analyze the efficiency of our approach on synthetic KBs (as in [9, 10]) and then compare the scalability of different preprocessing methods on the KGs of *MetaQA* and *WebQuestionsSP*. First, we perform experiments on KBs where the relational coalescing has no effect: the outdegree of each node is equal to the number of relation types and all edges originating from a node have different relation labels. We perform two experiments on such KBs. In the first one (Figure 5a), the number of relation types is fixed to $|\mathcal{R}| = 10$ and the number of entities varies from $|\mathcal{V}| = 10^2$ to $|\mathcal{V}| = 10^6$. In the second task (Figure 5b), the number of entities is fixed to $|\mathcal{V}| = 5000$ and the number of relations varies from $|\mathcal{R}| = 1$ to $|\mathcal{R}| = 10^3$. The single answer node is always two-hops away from the entities mentioned in the question. We compare **SQALER** (unrefined) against a GNN-based approach (**GRAFT-Net** [41]) and a key-value memory network (**KV-MemNN** [35]). The approaches are evaluated based on the queries per second at inference time with a mini-batch size of 1. The results show that increasing the number of entities has negligible impact on the performance of **SQALER**, whereas GRAFT-Net and the key-value memory network are limited to graphs with less than 10k nodes. This shows that, in large KGs like Freebase, the baselines would not be able to handle even a 2-hop neighborhood of the entities mentioned in the question (we refer the reader to Appendix E.5 for more details). Finally, from the results in Figure 5b, we see that the throughput of our approach decreases with the number of relation types. However, in practice, we can leverage the GPU to score the edges of the graph in parallel. This is why we observe only a minor drop in performance when the number of relation types grows from $|\mathcal{R}| = 1$ to $|\mathcal{R}| = 100$.

In order to assess the scalability of the proposed relational coalescing operation, we further compare commonly used preprocessing methods on the KG of *WebQuestionsSP*. We evaluate the time required to extract complete 2-hop neighborhoods of the entities mentioned in the question and the time to perform Personalized Page Rank (PPR) on such graphs. The results are shown in Figure 5c. Note that, at inference time, we can perform the coalescing only on the portion of the graph explored by the model, which makes **SQALER** much more efficient. At training time, the preprocessing is comparable to the 2-hop neighborhood extraction. Finally, Figure 5d shows the performance of the models with the respective preprocessing step at inference time on synthetic KBs with growing number of edges.

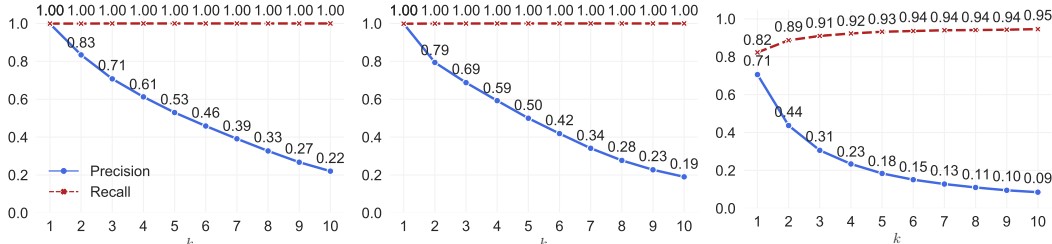

Figure 4: Precision and recall of the top $k$ sequences of relations on MetaQA 2 (left), MetaQA 3 (center) and WebQSP (right)

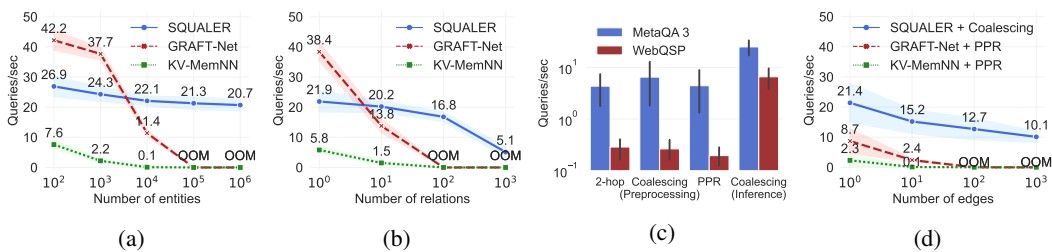

Figure 5: Inference time in queries/sec on synthetic KBs with increasing number of entities (a) and relation types (b). Time required by different preprocessing steps on the KG of *WebQuestionsSP* and *MetaQA* (c). Complete inference and preprocessing time on synthetic KBs with increasing number of edges (d). We set the queries/sec to 0 when the model runs out-of-memory (OOM).

## 4.4 Incomplete knowledge graphs

In order to evaluate the capability of our approach to cope with missing information in the knowledge graph, we performed two additional experiments. In the first experiment, we evaluated our approach (the **SQALER – GNN** variant) on *WebQuestionsSP* using incomplete knowledge graphs with only 50% of the original edges (**50% KG**). Then, following previous work [41, 42], we tried to mitigate the missing information using additional sources of external knowledge. In particular, for each question, we used the same text documents extracted from Wikipedia as done by Sun et al. [41] (**50% KG + Text**). In this experiment, the relation-level model is unaware of the additional source of knowledge, but the information from the text documents is infused into the edge-level GNN with the same strategy used in GRAFT-Net [41] (note that this makes the edge-level GNN-based model essentially equivalent to the full version of GRAFT-Net, with both KG and text support). We compare our approach against **GRAFT-Net** and **PullNet**, namely the two baselines designed for open-domain question answering with incomplete KGs and text documents.

The results of the experiments are reported in Table 3. We observe that, despite not being designed for incomplete KGs, **SQALER** outperforms the baselines on both experimental settings. This is not surprising, as **GRAFT-Net** relies on a simple heuristic process to construct question subgraphs and **PullNet** is constrained to follow the structure of the incomplete graph, because its iterative retrieval process can only expand nodes that are reachable from the set of anchor entities. This means that, in principle, any node retrieved by **PullNet**'s iterative process can also be reached by **SQALER**'s relation-level model. Similarly to the baselines, we note only a minor gain in performance when using the text documents as an additional source of information.

Table 3: Hits@1 on *WebQuestionsSP* with incomplete KGs (50% of the edges) and additional text

|  | 50% KG | 50% KG + Text |
| --- | --- | --- |
| **GRAFT-Net** [41] | 48.2 | 49.9 |
| **PullNet** [42] | 50.3 | 51.9 |
| **SQALER – GNN** | 53.5 | 55.2 |

# 5    Related work

Several lines of research in the past few years have focused on introducing deep learning approaches aimed at reasoning over structured knowledge. In particular, this paper is closely related to methods for learning to traverse KGs [14, 13, 24] and recent works on answering conjunctive queries using deep learning approaches [25, 16]. In this context, several KB and query embedding methods have been proposed [45]. Many KB embedding approaches support the same operation performed by our relation-level model, namely relation projection [10, 43, 25, 38]. Some KB embedding methods also explicitly learn to follow chains of relations and traverse KGs [24, 30, 13]. Notably, Query2Box [38] is a query embedding method that represents sets using box embeddings and the more recent beta embeddings [37] extend the framework to support a complete set of first-order logic operators. The main difference with our model is that these methods operate on vector space, whereas our approach is constrained on the graph structure and learns to traverse the KG while keeping the ability to scale to large graphs. Also, our method answers questions in natural language, while the above methods are primarily designed for query answering. Recently, Sun et al. [43] introduced EmQL, a query embedding method which has also been integrated in a question answering model.

Other lines of research on KBQA have focused on unsupervised semantic parsing [3, 2, 1] or on the introduction of supervised models, like graph neural networks (GNNs) designed for reasoning over knowledge graphs [41, 42, 48]. These approaches pose the KBQA problem as a node classification task. For this reason, they have been applied succesfully only on small query-dependent graphs. Cohen et al. [10] addressed the problem of creating a representation of a symbolic KB that enables building neural KB inference modules that are scalable enough to perform non-trivial inferences with large graphs. Another recent work [39] has explored using KG embeddings for question answering and handle incompleteness in the KG.

In our work, we combine relation projection with an edge-level GNN to address the KBQA problem. The same idea of combining GNNs with relational following was introduced in Gretel [11], which learns to complete natural paths in a graph given a path prefix. Also, our idea of accelerating GNNs by operating on a reduced graph representation has strong connections with graph coarsening and sparsification [32, 33, 6].

Methods based on reinforcement learning (RL) have also been proposed to perform multi-hop reasoning over knowledge graphs. Xiong et al. [46] proposed DeepPath, which relies on a policy-based agent that learns to reason over multi-hop paths by sampling relations at each step. Also, Das et al. [14] introduced MINERVA, a RL agent that learns how to navigate the graph conditioned only on an input entity and on a query. These approaches are designed for simple query answering and KB completion rather than KBQA. A main difference with our work is that **SQALER** samples multiple paths and employs an edge-level model to reach higher expressivity.

# 6    Conclusion

This paper introduced **SQALER**, a scalable approach to reasoning and question answering over KGs. Our method is expressive and can reach state-of-the-art performance on widely used and challenging datasets. Further, **SQALER** scales with the number of (distinct) relation types in the graph and can effectively handle large-scale knowledge graphs with millions of entities. Our empirical evaluation also showed that our approach can generalize compositionally and that it can be used to generate question-dependent subgraphs that strike a good trade-off between precision and recall.

Overall, our work proposes an improvement to existing KBQA technology which carries impact to several practical applications. Nevertheless, we remind that the deployment of such models needs to be done cautiously. KBQA replaces a mature technology (traditional KBs and query languages) with less understood methods. The underlying KB may be incomplete, contain misinformation or biases that could negatively affect the decisions of the learned model. We hope that our work will spur further research in this area and contribute to the development of reliable KBQA systems.

## Acknowledgments and Disclosure of Funding

Andreas Loukas would like to thank the Swiss National Science Foundation for supporting him in the context of the project "Deep Learning for Graph Structured Data", grant number PZ00P2 179981.

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
