# SQALER: Scaling Question Answering by Decoupling Multi-Hop and Logical Reasoning

# —

# Appendix

**Mattia Atzeni**
IBM Research, EPFL
Switzerland
atz@zurich.ibm.com

**Jasmina Bogojeska**
IBM Research
Switzerland
jbo@zurich.ibm.com

**Andreas Loukas**
EPFL
Switzerland
andreas.loukas@epfl.ch

## A  Formal definition of the coalesced representation

Given a knowledge graph $\mathcal{G} = (\mathcal{V}, \mathcal{R}, \mathcal{E})$ and a set of entities $\mathcal{V}_Q$, we can provide an alternative recursive definition of $reach_{\mathcal{G}}(\mathcal{V}_Q, R)$ as:

$$reach_{\mathcal{G}}(\mathcal{V}_Q, (r_1, r_2, \ldots, r_{|R|})) = \begin{cases} \mathcal{V}_Q & \text{if } |R| = 0 \\ reach_{\mathcal{G}}(\mathcal{V}'_Q, (r_2, \ldots, r_{|R|})) & \text{if } \mathcal{V}_Q \xrightarrow{r_1} \mathcal{V}'_Q \\ \emptyset & \text{otherwise} \end{cases}$$

where $\mathcal{V}'_Q$ is the set of nodes reachable from $\mathcal{V}_Q$ by an $r_1$ relation.

Then, we can define the coalesced representation $\tilde{\mathcal{G}}_Q = (\tilde{\mathcal{V}}_Q, \tilde{\mathcal{R}}_Q, \tilde{\mathcal{E}}_Q)$ as follows:

- $\tilde{\mathcal{V}}_Q = \{reach_{\mathcal{G}}(\mathcal{V}_Q, R) \mid R \in \mathcal{R}^*\}$ are the nodes $R$-reachable from $\mathcal{V}_Q$ by $R \in \mathcal{R}^*$ ($*$ is the Kleene star);

- $\tilde{\mathcal{R}}_Q = \mathcal{R} \cup \{\texttt{self}\}$ is the original set of relations augmented with the self-loop relation type $\texttt{self}$, which denotes the empty sequence $\texttt{self} \in \mathcal{R}^*$;

- edge $\mathcal{V}_i \xrightarrow{r} \mathcal{V}_j$ belongs to $\tilde{\mathcal{E}}_Q$ if and only if $\mathcal{V}_j = reach_{\mathcal{G}}(\mathcal{V}_i, r)$, with $r \in \tilde{\mathcal{R}}_Q$.

Intuitively, this operation can be seen as coalescing relations in the original knowledge graph $\mathcal{G}$ and adding self loops. In practice, we do not need to compute all the nodes in $\tilde{\mathcal{G}}_Q$ but only edge labels.

## B  Computational Complexity

The knowledge seeking procedure described in Section 2.1 applies a search algorithm over the graph $\tilde{\mathcal{G}}_Q$ to obtain the most likely set of relation sequences originating from $\mathcal{V}_Q$. The exact knowledge seeking procedure adopted in our experiments is based on the beam search algorithm and is detailed in Algorithm 1. The algorithm is designed to scale with the number of relation types in the original knowledge graph, which is usually much smaller than the number of edges (facts) or nodes (entities). In this section, we describe the algorithm in more details and we provide an extensive analysis of the computational complexity of our approach.

**Overview of the knowledge seeking procedure.**    At each iteration, Algorithm 1 updates a set $\mathcal{B}_t$ containing triples of the form $(\mathcal{V}_t, R_t, w_t)$. We denote with $\mathcal{V}_t = reach_{\mathcal{G}}(\mathcal{V}_Q, R_t)$ the set of nodes reachable from $\mathcal{V}_Q$ by following $R_t$, whereas $R_t$ represents a relation sequence constructed iteratively

35th Conference on Neural Information Processing Systems (NeurIPS 2021).

by applying the relation-level model on edges of $\tilde{\mathcal{G}}_Q$ up to time step $t$. The last element of the tuples $w_t$ is the total accumulated negative log-likelihood of $R_t$, computed as explained in Section 2.1. At the beginning of the algorithm, $\mathcal{V}_1 = \mathcal{V}_Q$ is the set of entities mentioned in the natural language question, $R_1 = \texttt{self}$ is the empty relation sequence and we set the initial negative log-likelihood $w_1 = 0$. The algorithm receives as input a parameter $\beta$ which specifies the *beam width*, namely the number of relation sequences that are expanded at each iteration. At time step $t$, we compute the set $\tilde{\mathcal{E}}_t$ of all edges originating from $\mathcal{V}_t$ in $\tilde{\mathcal{G}}_Q$. Then, the relation sequences $R_t$ are expanded with the relation types labeling edges in $\tilde{\mathcal{E}}_t$. The likelihood of the new relation sequences is calculated based on $w_t$ and the likelihood assigned by the relation-level model to the relation type appended to $R_t$. At the end of each iteration, the function $\min(\mathcal{B}_{t+1}, \beta)$ in Algorithm 1 retains for the next time step only the $\beta$ tuples $(\mathcal{V}_{t+1}, R_{t+1}, w_{t+1}) \in \mathcal{B}_{t+1}$ with the minimum negative log-likelihood $w_{t+1}$. Note that, in Algorithm 1, relation sequences ending with the $\texttt{self}$ relation type are not expanded after the first time step. As explained in Section 3, indeed, the $\texttt{self}$ relation type is used to signal both the start and the end of the decoding.

**Time complexity.** At time step $t$, for each triple $(\mathcal{V}_t, R_t, w_t) \in \mathcal{B}_t$, the algorithm computes $\phi$ for all edges $\tilde{\mathcal{E}}_t$ originating from $\mathcal{V}_t$. This means that the relation-level model described in Section 3 is queried $|\mathcal{B}_t| \cdot |\tilde{\mathcal{E}}_t|$ times at iteration $t$. Note that we do not need to compute the likelihood $\phi(\mathcal{V}_i \xrightarrow{r} \mathcal{V}_j)$ for all edges $\mathcal{V}_i \xrightarrow{r} \mathcal{V}_j$ in $\tilde{\mathcal{E}}_Q$. Let $d_{max}^+(\tilde{\mathcal{G}}_Q)$ be the maximum outdegree of nodes in $\tilde{\mathcal{G}}_Q$. At time step $t$, the size of the set $\mathcal{B}_{t+1}$ is restricted to $\beta$ for the next iteration by the operation $\min(\mathcal{B}_{t+1}, \beta)$. Since $|\mathcal{B}_t|$ is bounded by $\beta$ and $|\tilde{\mathcal{E}}_t|$ is bounded by $d_{max}^+(\tilde{\mathcal{G}}_Q)$, at any iteration, the relation-level model is queried at most $\beta \cdot d_{max}^+(\tilde{\mathcal{G}}_Q)$ times. Each of such queries takes constant time. The function $\min(\mathcal{B}_{t+1}, \beta)$ selects the $\beta$ tuples in $\mathcal{B}_{t+1}$ with the smallest negative log-likelihood. This can be done on average in $\mathcal{O}(|\mathcal{B}_{t+1}|)$ time. At iteration $t$, the set $\mathcal{B}_{t+1}$ is initialized as the empty set and updated

---

**Algorithm 1:** Knowledge Seeking

**Input** : a coalesced knowledge graph $\tilde{\mathcal{G}}_Q$; a set of starting entities $\mathcal{V}_Q$; the beam width $\beta$; the maximum number of iterations $\tau_{max}$; and the number of relation sequences to be returned $k \leq \beta$

**Output :** A set of $k$ tuples of the form $(\tilde{\mathcal{A}}_Q, R, w)$, representing the $k$ most likely candidate answers $\tilde{\mathcal{A}}_Q$, the sequence of relations $R$ to reach $\tilde{\mathcal{A}}_Q$, and the negative log-likelihood $w$ of $R$

$t \leftarrow 1$
$\mathcal{B}_t \leftarrow \{(\mathcal{V}_Q, \texttt{self}, 0)\}$

**repeat**
    $\mathcal{B}_{t+1} \leftarrow \emptyset$
    **for** $(\mathcal{V}_t, R_t, w_t) \in \mathcal{B}_t$ **do**
        **if** $R_t = (r_0, \ldots, \texttt{self})$ **and** $t > 1$ **then**
            $\mathcal{B}_{t+1} \leftarrow \mathcal{B}_{t+1} \cup \{(\mathcal{V}_t, R_t, w_t)\}$
        **else**
            $\tilde{\mathcal{E}}_t \leftarrow \{\mathcal{V}_t \xrightarrow{r_t} \mathcal{V}_{t+1} \in \tilde{\mathcal{E}}_Q\}$
            **for** $\mathcal{V}_t \xrightarrow{r_t} \mathcal{V}_{t+1} \in \tilde{\mathcal{E}}_t$ **do**
                $R_{t+1} = (R_t, r_t)$
                $w_{t+1} \leftarrow w_t - \log \phi(\mathcal{V}_t \xrightarrow{r_t} \mathcal{V}_{t+1})$
                $\mathcal{B}_{t+1} \leftarrow \mathcal{B}_{t+1} \cup \{(\mathcal{V}_{t+1}, R_{t+1}, w_{t+1})\}$
            **end**
        **end**
    **end**
    $\mathcal{B}_{t+1} \leftarrow \min(\mathcal{B}_{t+1}, \beta)$
    $t \leftarrow t + 1$
**until** $\mathcal{B}_t = \mathcal{B}_{t-1}$ **or** $t > \tau_{max}$

**return** $\min(\mathcal{B}_t, k)$

---

by adding at most $\beta \cdot d_{max}^+(\tilde{\mathcal{G}}_Q)$ tuples (one element for each query to the relation-level model). Therefore, the expected time complexity of the function $\min(\mathcal{B}_{t+1}, \beta)$ is $\mathcal{O}(\beta \cdot d_{max}^+(\tilde{\mathcal{G}}_Q))$. Now, note that by the definition of $\tilde{\mathcal{G}}_Q$, we have $d_{max}^+(\tilde{\mathcal{G}}_Q) \leq |\tilde{\mathcal{R}}_Q| = |\mathcal{R}| + 1$. Hence, the number of queries to the relation-level model is bounded by $\beta \cdot (|\mathcal{R}| + 1)$ and the time complexity of $\min(\mathcal{B}_{t+1}, \beta)$ is also $\mathcal{O}(\beta \cdot |\mathcal{R}|)$. The maximum depth reached by the knowledge seeking procedure starting from $\mathcal{V}_Q$ is bounded by $\tau_{max}$, because Algorithm 1 performs at most $\tau_{max}$ iterations of the main outer loop. The final step $\min(\mathcal{B}_t, k)$ selects the $k$ most likely tuples and can be run on average in $\mathcal{O}(\beta)$ time. This yields a final computational complexity of

$$\mathcal{O}(\tau_{max} \cdot \beta \cdot |\mathcal{R}|) = \mathcal{O}(|\mathcal{R}|).$$

Note that $\tau_{max}$ and $\beta$ are constant parameters of the algorithm and are usually small. In our experiments, we set $\tau_{max} = 3$ for MetaQA 3 and $\tau_{max} = 2$ for MetaQA 2 and WebQSP. We set the beam width $\beta = 10$, obtaining only minor improvements with respect to a greedy search with $\beta = 1$. Therefore, we obtain that that time complexity of the knowledge seeking procedure scales linearly with the number of relation types and does not depend on the number of nodes or edges in $\mathcal{G}$.

**Space complexity.** For each iteration $t$, Algorithm 1 constructs $\mathcal{B}_{t+1}$ by analyzing all edges originating from each node $\mathcal{V}_t$ stored in the tuples $(\mathcal{V}_t, R_t, w_t) \in \mathcal{B}_t$. From the considerations reported above, the size of $\mathcal{B}_{t+1}$ is $\mathcal{O}(\beta \cdot |\mathcal{R}|)$. Although for notational convenience we are representing $\mathcal{B}_t$ as a set of triples, in practice we can avoid storing intermediate nodes $\mathcal{V}_t$ and construct the set of candidate answers by following $R_t$ at the final iteration. Therefore, we only need to store relation sequences $R_t$ and their negative log-likelihood $w_t$. Each tuple requires $\mathcal{O}(\tau_{max})$ space, as $|R_t|$ is bounded by $\tau_{max}$. The space complexity of the algorithm is thus $\mathcal{O}(\tau_{max} \cdot \beta \cdot |\mathcal{R}|)$.

## C   Expressive Power

As mentioned in Section 2.3, the approach described in this paper can be used to answer any valid *existential positive first order query* on a knowledge graph $\mathcal{G}$. In order to prove this, we first consider the simpler class of *conjunctive queries*. We will show a result similar to Proposition 1 for conjunctive queries, and then we will extend this result to the wider class of EPFO queries.

### C.1   Conjunctive Queries

Given a knowledge graph $\mathcal{G} = (\mathcal{V}, \mathcal{R}, \mathcal{E})$ and a non-empty set of nodes $\mathcal{V}_Q \subseteq \mathcal{V}$, a conjunctive query on $\mathcal{G}$ is a query involving only existential quantification and conjunction:

$$\mathsf{Q}[\mathsf{V}_?] = \mathsf{V}_?.\exists \mathsf{V}_1, \ldots, \mathsf{V}_m : \mathsf{e}_1 \wedge \mathsf{e}_2 \wedge \cdots \wedge \mathsf{e}_{|\mathsf{Q}|},$$

such that each literal $\mathsf{e}_i$ is an atomic formula of the form $r(\mathsf{V}, \mathsf{V}')$, where $\mathsf{V} \in \mathcal{V}_Q \cup \{\mathsf{V}_1, \ldots, \mathsf{V}_m\}$, $\mathsf{V}' \in \{\mathsf{V}_?, \mathsf{V}_1, \ldots, \mathsf{V}_m\}$, $\mathsf{V} \neq \mathsf{V}'$, and $r(\mathsf{V}, \mathsf{V}')$ is satisfied if and only if $\mathsf{V} \xrightarrow{r} \mathsf{V}'$, $r \in \mathcal{R}$.

In general, for any query $\mathsf{Q}$, we can define its *dependency graph* as the graph with nodes $\mathcal{V}_Q \cup \{\mathsf{V}_?, \mathsf{V}_1, \ldots, \mathsf{V}_m\}$. The edges of the graph are the literals $\{\mathsf{e}_1, \ldots, \mathsf{e}_{|\mathsf{Q}|}\}$, as each literal is of the form $r(\mathsf{V}, \mathsf{V}')$ and defines an edge between $\mathsf{V}$ and $\mathsf{V}'$ [25]. Figure 1 shows an example of the dependency graph of a conjunctive query.

We say that a query is *valid* if its dependency graph is a directed acyclic graph (DAG), with $\mathcal{V}_Q$ as the source nodes and the target variable $\mathsf{V}_?$ as the unique sink node. In the following, we will always consider valid queries, as this ensures that the query has no redundancies or contradictions.

**Lemma 1.** *Let $\mathcal{G} = (\mathcal{V}, \mathcal{R}, \mathcal{E})$ be a knowledge graph and $\mathsf{Q}$ be a valid conjunctive query on $\mathcal{G}$. Then, there exists a sequence of relations $R^\star \in \mathcal{R}^*$ such that:*

$$\mathcal{A}_Q \subseteq reach_{\mathcal{G}}(\mathcal{V}_Q, R^\star),$$

*where $\mathcal{A}_Q = \{v \in \mathcal{V} \mid \mathsf{Q}[v] = \mathsf{True}\}$ is the denotation set of $\mathsf{Q}$, namely the entities satisfying $\mathsf{Q}$.*

*Proof.* We proceed by induction on the number of literals $|\mathsf{Q}|$.

*Base case.* Assume $|Q| = 1$. Then, since Q is valid, the query is of the form:

$$Q[V_?] = V_?.r(v, V_?),$$

with $\{v\} = \mathcal{V}_Q$. We have:

$$\begin{aligned}
\mathcal{A}_Q &= \{v' \in \mathcal{V} \mid Q[v'] = \mathsf{True}\} \\
&= \{v' \in \mathcal{V} \mid v \xrightarrow{r} v'\} \\
&= reach_\mathcal{G}(\mathcal{V}_Q, r).
\end{aligned}$$

Hence the sequence with only relation $r$ is sufficient to generate the set of correct answers $\mathcal{A}_Q$.

*Inductive step.* Let Q be a conjunctive query of the form:

$$Q[V_?] = V_?.\exists V_1, \ldots, V_m : e_1 \wedge e_2 \wedge \cdots \wedge e_{|Q|}.$$

Assume that there exists a sequence of relations $R^\star \in \mathcal{R}^*$, such that:

$$\mathcal{A}_Q = \{v \in \mathcal{V} \mid Q[v] = \mathsf{True}\} \subseteq reach_\mathcal{G}(\mathcal{V}_Q, R^\star).$$

Consider a query Q' constructed by adding a literal $e_{|Q|+1}$ to Q, and let $\mathcal{A}'_Q$ be the denotation set of Q', namely the set of nodes satisfying Q'. The conjunctive query Q' may or may not have the same target variable of Q.

If Q' shares the same target variable of Q, then Q' is of the form:

$$Q'[V_?] = V_?.\exists V_1, \ldots, V_m : e_1 \wedge e_2 \wedge \cdots \wedge e_{|Q|} \wedge e_{|Q+1|}.$$

Note that:

$$\begin{aligned}
\mathcal{A}'_Q &= \{v \in \mathcal{V} \mid Q'[v] = \mathsf{True}\} \\
&\subseteq \{v \in \mathcal{V} \mid Q[v] = \mathsf{True}\} \\
&\subseteq reach_\mathcal{G}(\mathcal{V}_Q, R^\star).
\end{aligned}$$

Hence, if Q and Q' share the same target variable, the same sequence of relations that generates candidate answers for Q can be used to generate candidate answers for Q'.

If Q and Q' do not have the same target variable, then we can write Q' as:

$$Q'[V'_?] = V'_?.\exists V_?, V_1, \ldots, V_m : e_1 \wedge e_2 \wedge \cdots \wedge e_{|Q|} \wedge e_{|Q+1|}.$$

Since Q' is a *valid* conjunctive query on $\mathcal{G}$, $e_{|Q+1|}$ is of the form $r(V_?, V'_?)$. Then we have that:

$$\begin{aligned}
\mathcal{A}'_Q &= \{v' \in \mathcal{V} \mid Q'[v'] = \mathsf{True}\} \\
&= \{v' \in \mathcal{V} \mid \exists V_?, V_1, \ldots, V_m : e_1 \wedge e_2 \wedge \cdots \wedge e_{|Q|} \wedge V_? \xrightarrow{r} v'\} \\
&= \{v' \in \mathcal{V} \mid \exists v \in \mathcal{A}_Q : v \xrightarrow{r} v'\} \\
&= reach_\mathcal{G}(\mathcal{A}_Q, r) \\
&\subseteq reach_\mathcal{G}(reach_\mathcal{G}(\mathcal{V}_Q, R^\star), r) \\
&= reach_\mathcal{G}(\mathcal{V}_Q, (R^\star, r)).
\end{aligned}$$

Therefore, the sequence $(R^\star, r)$ can be used to generate the answers to Q'. $\qquad\square$

## C.2 Existential Positive First-Order Queries

Any EPFO query can be expressed in disjunctive normal form (DNF), namely a disjunction of one or more conjunctions:

$$Q[V_?] = V_?.\exists V_1, \ldots, V_m : C_1 \vee C_2 \vee \cdots \vee C_{n_\vee + 1},$$

such that:

- each $C_i$ is a conjunction of literals of the form $C_i = e_{i1} \wedge e_{i2} \wedge \cdots \wedge e_{i|C_i|}$
- each literal $e_{ij}$ is an atomic formula of the form $r(V, V')$, where $V \in \mathcal{V}_Q \cup \{V_1, \ldots, V_m\}$, $V' \in \{V_?, V_1, \ldots, V_m\}$, $V \neq V'$, and $r(V, V') = \mathsf{True}$ if and only if $V \xrightarrow{r} V'$, $r \in \mathcal{R}$.

As above, we assume that the Q is a *valid* query on $\mathcal{G}$, namely all $C_i$ are valid conjunctive queries. As shown in Figure 1, we can represent any EPFO query Q with a computation graph containing the operations that are required to answer Q. Specifically, each atomic formula can be represented as a relation projection, whereas conjunctions and disjunctions can be represented as intersection and union operations respectively.

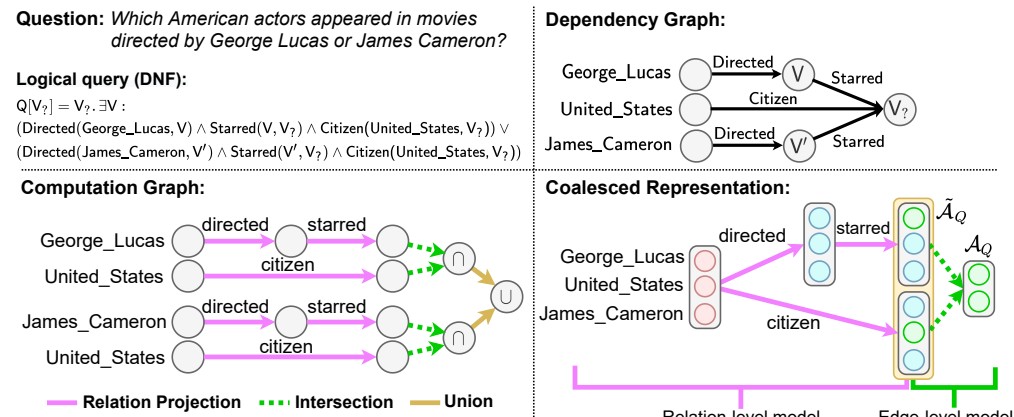

Figure 1: Example of a natural language question and the corresponding EPFO query expressed in DNF (top left); the dependency graph of the EPFO query (top right); the computation needed to answer the query in the original KG (bottom left); and the computation performed by our approach in the coalesced representation (bottom right). Note that, for completeness, we represent two paths in the coalesced representation, but only one is sufficient.

**Proof of Proposition 1.** We assume that $Q$ is expressed in disjunctive normal form and we denote with $n_\vee$ the number of disjunction ($\vee$) operators in $Q$. We proceed by induction on $n_\vee$.

*Base case.* Assume $n_\vee = 0$. Then, $Q$ is a conjunctive query, and by Lemma 1, there exists $R^\star \in \mathcal{R}^*$ such that:

$$\mathcal{A}_Q = \{v \in \mathcal{V} \mid Q[v] = \mathsf{True}\} \subseteq reach_\mathcal{G}(\mathcal{V}_Q, R^\star).$$

*Inductive step.* Let $Q$ be an EPFO query in DNF:

$$Q[\mathsf{V}_?] = \mathsf{V}_?.\exists \mathsf{V}_1, \ldots, \mathsf{V}_m : \mathsf{C}_1 \vee \mathsf{C}_2 \vee \cdots \vee \mathsf{C}_{n_\vee+1}.$$

Consider the subquery $Q'$ consisting of the conjunction terms $\mathsf{C}_1 \vee \mathsf{C}_2 \vee \cdots \vee \mathsf{C}_{n_\vee}$ and assume that there exist $k \le n_\vee$ sequences of relations $R_i^\star$ such that:

$$\mathcal{A}'_Q = \{v \in \mathcal{V} \mid Q'[v] = \mathsf{True}\} \subseteq \bigcup_{i=1}^{k} reach_\mathcal{G}(\mathcal{V}_Q, R_i^\star).$$

Note that $\mathsf{C}_{n_\vee+1}$ is a valid conjunctive query and by Lemma 1 there exists $R_{k+1}^\star \in \mathcal{R}^*$ such that:

$$\{v \in \mathcal{V} \mid \mathsf{C}_{n_\vee+1}[v]\} \subseteq reach_\mathcal{G}(\mathcal{V}_Q, R_{k+1}^\star).$$

Then, it holds that:

$$\begin{aligned}
\mathcal{A}_Q &= \{v \in \mathcal{V} \mid Q[v] = \mathsf{True}\} \\
&= \{v \in \mathcal{V} \mid Q'[v] \vee \mathsf{C}_{n_\vee+1}[v]\} \\
&= \mathcal{A}'_Q \cup \{v \in \mathcal{V} \mid \mathsf{C}_{n_\vee+1}[v]\} \\
&\subseteq \bigcup_{i=1}^{k} reach_\mathcal{G}(\mathcal{V}_Q, R_i^\star) \cup \{v \in \mathcal{V} \mid \mathsf{C}_{n_\vee+1}[v]\} \\
&\subseteq \bigcup_{i=1}^{k+1} reach_\mathcal{G}(\mathcal{V}_Q, R_i^\star).
\end{aligned}$$

$\square$

# D   Training strategies

In this section we describe the training strategies that we used to optimize the parameters of our relation-level model and improve generalization performance.

**Supervision.** For the experiments on KBQA, we assume that we only have access to pairs of questions and answers, i.e. the actual inferential chain leading from the question to the answer is latent. Therefore, we resort to weak supervision to train the model. Since at training time the set $\mathcal{A}_Q$ is known, we can compute all relation sequences $R^\star$, such that $\tilde{\mathcal{A}}_Q = reach_{\mathcal{G}}(\mathcal{V}_Q, R^\star)$ is the smallest reachable superset of $\mathcal{V}_Q$. If the smallest reachable superset of $\mathcal{A}_Q$ is not unique, all relation sequences leading to any superset of $\mathcal{A}_Q$ of minimum cardinality are considered. Note that the set of all possible relation sequences of a given length originating from $\mathcal{V}_Q$ in $\tilde{\mathcal{G}}_Q$ is much smaller than the set of all possible paths starting from nodes in $\mathcal{V}_Q$ in $\mathcal{G}$, as shown in Appendix E.5. Since the *CFQ* dataset contains boolean questions (where the answer is not a set of entities), for the experiment on compositional generalization we use the logical parsing provided in the dataset to compute the correct sequences of relations. We assume these sequences of relations are stored in such a way that the set of relations exiting from the a node in $\tilde{\mathcal{G}}_Q$ can be accessed efficiently in constant time. Then, at any decoding time step $t$, an edge is labeled as positive if and only if it belongs to a sequence of relations leading to $\tilde{\mathcal{A}}_Q$. The model is then trained using teacher forcing, namely we feed into the decoder relation sequences leading from $\mathcal{V}_Q$ to $\tilde{\mathcal{A}}_Q$. We do not have multiple decoding time steps at training time, as the whole sequence is provided at once, and relation types are appropriately masked so that they cannot attend to items in future positions.

**Path dropout.** Previous work [41] has shown that randomly removing facts from the knowledge base at training time can be beneficial for generalization. Inspired by such insight, we employ a similar technique to enhance the performance of our model. Specifically, in the first epochs, we randomly remove paths that are not labeled as correct with probability $p_{drop}$, in order to make the problem easier for the model. This probability is the linearly decreased to 0 during training. We set the initial $p_{drop}$ to 0.5 and we gradually lower it to 0 until half of the training epochs have been run.

**Pretraining and fine tuning.** For the experiments on *WebQuestionsSP* and *CFQ*, we found it beneficial to pretrain our model in order to incorporate knowledge from Freebase into the layers of the decoder. Specifically, we sampled a total of 500k 1-hop or 2-hop paths and we trained the model to predict the sequence of relations connecting two nodes, given the embeddings of the source node and the target node of the path. In order to do this, we replace the encoder with a simple 2-layer feed-forward network, with a ReLU non-linearity. This network receives as input two 100-dimensional embeddings for the source and target nodes of the path, and maps them to a $d_{model}$-dimensional representation. This representation is then fed into the decoder to predict the relations connecting the two nodes. We use a concatenation of 50-dimensional random and 50-dimensional pretrained TransE [8] embeddings [26] to represent the entities in the KG. Moreover, on *WebQuestionsSP* we observed that it was helpful to fine-tune BERT in order to produce better representations of the relations in the knowledge graph. The same BERT model is still used to encode both the questions and relation types.

# E Experimental Details

## E.1 Datasets

**KBQA datasets.** We performed our experiments on KBQA on two widely adopted datasets, namely *MetaQA* [50] and *WebQuestionsSP* [49]. We provide below a detailed description of each one.

- **MetaQA**[1] [50] is a multi-hop question answering dataset including 400K question-answer pairs. Questions are answerable using the WikiMovies knowledge base. The dataset includes 1-hop, 2-hop and 3-hop questions. It is provided under the Creative Commons Public License Attribution 3.0 Unported[2]. We evaluated our approach on 2-hop (**MetaQA 2**) and 3-hop (**MetaQA 3**) questions.

- **WebQuestionsSP** [49] comprises 4737 questions over a subset of Freebase, which is provided under the CC BY 2.5 license[3]. The questions in this dataset are answerable by performing relational following for up to two hops and an optional relational filtering operation on the result.

---

[1]`https://github.com/yuyuz/MetaQA`
[2]`https://creativecommons.org/licenses/by/3.0/`
[3]`https://creativecommons.org/licenses/by/2.5/`

Table 1 shows the number of questions in the training, development and test splits of each dataset. We use the same splits as in [41]. Table 3 reports instead the number of triples (edges), entities (nodes) and relations in the KGs used in our experiments.

Table 1: Number of questions in the training, development and test sets

|           | Train   | Dev    | Test   |
|-----------|---------|--------|--------|
| **MetaQA 2** | 118 980 | 14 872 | 14 872 |
| **MetaQA 3** | 114 196 | 14 274 | 14 274 |
| **WebQSP**   | 2 848   | 250    | 1 639  |

Table 2: Size of the knowledge graphs used for *MetaQA* and *WebQuestionsSP*

|          | Triples    | Entities  | Relations |
|----------|------------|-----------|-----------|
| **MetaQA**  | 392 906    | 43 230    | 18        |
| **WebQSP**  | 23 587 078 | 7 448 928 | 575       |

**Compositional generalization.** Our experiments on compositional generalization rely on the *Compositional Freebase Questions* (*CFQ*) dataset. It includes a total of 239 357 English question-answer pairs that are answerable using the public Freebase data [21]. *CFQ* is released under the CC-BY-4.0 license[4] provides train-test splits designed to measure the compositional generalization ability of a machine-learning model. Each question is composed of primitive elements (*atoms*), which include entity mentions, predicates and question patterns. These atoms can be combined in different ways (*compounds*) to instantiate the specific examples in the dataset. The train-test splits are designed with the twofold goal of:

1. *minimizing atom divergence*: the atoms present in the test set are also included in the training set and their distribution in the test set is as similar as possible to their distribution in the test set;

2. *maximizing compound divergence*: the distribution of compounds in the test set is as different as possible from their distribution in the training set.

The dataset provides three different splits (**MCD1**, **MCD2**, **MCD3**), with *maximum compound divergence* (MCD) and low atom divergence. For each question, both a logical parsing and the expected answers are included. Hence, *CFQ* can be used both for semantic parsing and end-to-end question answering.

### E.2 Baselines

**KBQA Baselines.** On *WebQuestionsSP* and *MetaQA*, we compared our approach against the following baselines.

- **KV-MemNN** is a key-value memory network [35] that makes use of a memory of key-value pairs to store the triples from the KG. Keys are joint representation of the subject and relation of each triple, whereas the objects of the triples are used as the corresponding values.

- **ReifKB** [10] uses a compact encoding for representing symbolic KGs, called a sparse-matrix reified KG, which can be distributed across multiple GPUs, allowing efficient symbolic reasoning.

- **GRAFT-Net** [41] is a graph neural network designed to reason over question-specific subgraphs. The message-passing scheme is conditioned on the input question and takes inspiration from personalized page rank to perform a directed propagation of the messages starting from the entities mentioned in the question.

- **PullNet** [42] builds on top of GRAFT-Net and improves the quality of the question-specific subgraphs with an iterative process based on a learned classifier. This classifier selects which node should be expanded at each iteration and it is a further GNN with the same architecture as GRAFT-Net.

---

[4]https://creativecommons.org/licenses/by/4.0/

- **EmbedKGQA** [39] uses KG embeddings for multi-hop question answering.
- **EmQL** [43] relies on a query embedding method that combines a count-min sketch representation for entity sets with logical operations implemented via neural retrieval over embedded KG triples.

*CFQ* **Baselines.** For the experiment on compositional generalization, we compare to the best-performing baselines in *CFQ*'s public leaderboard[5]. These baselines are all designed for semantic parsing and are encoder-decoder architectures trained to output a formal query given a natural language question. Keysers et al. [28] evaluated the compositional generalization capabilities of three sequence-to-sequence models, namely one based on LSTMs [27] equipped with an attention mechanism [5] (**LSTM + Attention**), a **Transformer** [44] and a **Universal Transformer** [17]. Furrer et al. [20] conducted a study that assessed the performance of three more models. The **Evolved Transformer** [40] is a variation of the Transformer discovered with an evolutionary neural architecture search seeded with the original model of Vaswani et al. [44]. The *Text-to-Text Transfer Transformer* (T5) [36] is a model pre-trained to treat every task as a text-to-text problem. Furrer et al. [20] fine-tuned all variants, including the largest one with 11 billion parameters (**T5-11B**). Following the technique introduced by Guo et al. [22], Furrer et al. [20] further implemented the variant **T5-11B-mod**, which learns to predict an intermediate representation of the SPARQL query which is closer to the formulation of the questions in natural language. Finally, Guo et al. [23] introduced the *Hierarchical Poset Decoding* (**HPD**), which enforces partial permutation invariance, thus taking into account semantics and capturing higher-level compositionality.

### E.3 Hyperparameters and Reproducibility

We train the relation-level model for 300 epochs on both datasets. We use a mini-batch size of 128 for *MetaQA* and 32 for *WebQuestionsSP*. We set the dimension of the embeddings to $d_{model} = 768$, as we use 12 attention heads applied to tensors of size 64. We optimize the model using the AdamW optimizer [31], with weight decay of $10^{-3}$. The initial learning rate is set to $10^{-4}$ for *MetaQA* and $5 \cdot 10^{-6}$ for *WebQuestionsSP*. We apply dropout regularization, with probability 0.1 on both the encoder and the decoder layers. We use a beam width $\beta = 10$ for the knowledge seeking algorithm described in Appendix B.

BERT is fine-tuned for *WebQuestionsSP*, whereas the weights are kept fixed for *MetaQA*. For experiments on *WebQuestionsSP*, we found beneficial to pretrain our model in order to incorporate knowledge from Freebase into the layers of the decoder, as explained in Appendix D. Specifically, we sampled a total of 500k 1-hop or 2-hop paths and we trained the model to predict the sequence of relations connecting two nodes, given the embeddings of the source node and the target node of the path.

For the GCN-based edge-level model, we used the same implementation and hyperparameters of the version publicly available at: `https://github.com/OceanskySun/GraftNet`. All experiments are performed on a NVIDIA Tesla V100 GPU with 16 GB of memory.

### E.4 Discussion and qualitative examples

In our experiments on KBQA, for each model, we selected the entity $v^\star \in \mathcal{V}$ with the highest likelihood to be a correct answer. The answer to the question is considered correct if $v^\star \in \mathcal{A}_Q$. For the unrefined **SQALER** model, we report the expected performance selecting $v^\star$ uniformly at random from $\tilde{\mathcal{A}}_Q$.

The high performance of the unrefined model on *MetaQA* confirms our hypothesis that the relation-level model applied on the coalesced representation is sufficient for tasks such as multi-hop question answering. On *WebQuestionsSP*, the edge-level model is needed because relation projection is not sufficient to answer some of the questions in the dataset. Figure 2 shows some examples of the relation sequences predicted by our model on the test set of *WebQuestionsSP*. For examples *(a)* and *(b)*, the relation-level model is sufficient, as the set of candidates $\tilde{\mathcal{A}}_Q$ is the same as the set of the actual answers $\mathcal{A}_Q$. However, examples *(c)* and *(d)* demonstrate the need for an edge-level model, as following a sequence of relations is not always sufficient to obtain the correct answer. Note that the edge-level model is applied on a 1-hop neighborhood expansion of the graphs depicted in Figure 2

---

[5] `https://github.com/google-research/google-research/tree/master/cfq`

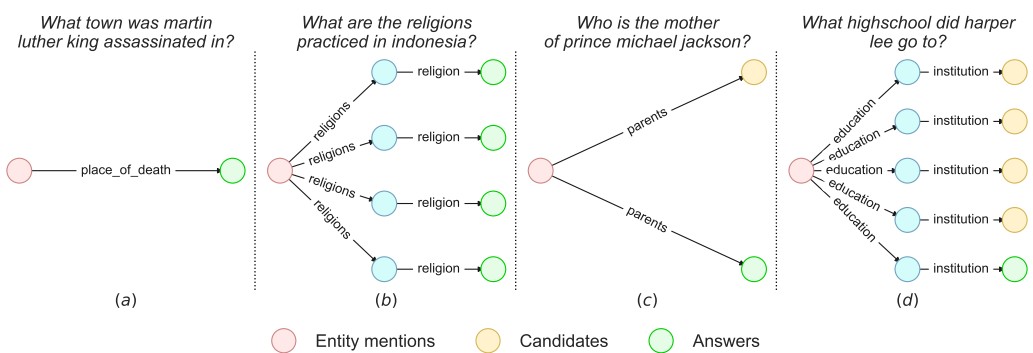

Figure 2: Example of relation sequences predicted by the unrefined relation-level model on the test set of *WebQuestionsSP*

and constrained to select an answer among the candidates $\tilde{\mathcal{A}}_Q$. Figure 2 also shows that the answer paths for 2-hop questions in *WebQuestionsSP* always contain compound value type (CVT) entities in the middle (depicted with cyan nodes in the image). These are special entity types that are used in *Freebase* to describe $n$-ary relationships between entities. EmQL uses different encodings for CVT nodes and the real entities, while **SQALER** does not depend on the KG specifics.

## E.5    Analysis of relational coalescing

In order to answer a question that requires multi-hop reasoning over a KG correctly, one should ideally either consider the full KG or a complete subgraph consisting of all possible multi-hop neighbors of the entities mentioned in the question. However, such subgraphs can be very large, as shown in Table 3. The subgraphs we analyzed include 3-hop neighbors for **MetaQA 3** and both 1 and 2-hop neighbors of the entities mentioned in the question for **WebQSP**. The average number of nodes for MetaQA 3 exceeds 10k and for the larger Freebase KG there are question subgraphs with millions of nodes. This makes impractical to perform KBQA on complete subgraphs with models that scale with the number of edges or nodes in the graph.

Table 3: Size of the subgraphs including all neighbors of the entities mentioned in a question

|  | MetaQA 3 | WebQSP |
|---|---|---|
| **Mean nodes** | 8.6k | 36k |
| **Max nodes** | 30k | 1.6M |
| **Mean facts** | 49k | 211k |
| **Max facts** | 230k | 9.5M |

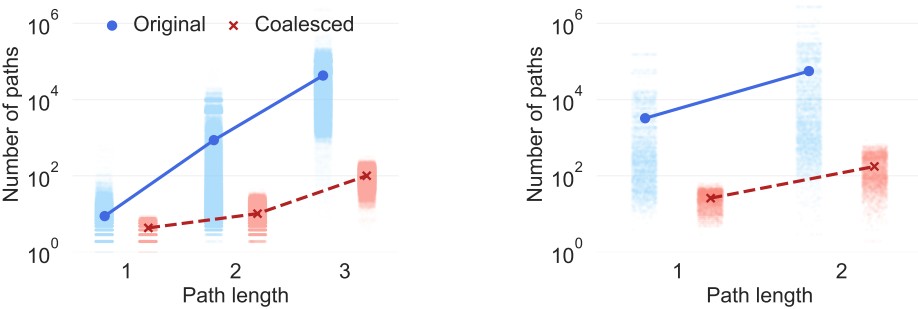

Figure 3: Number of paths by path length in the original and coalesced graphs for *MetaQA* (right) and *WebQuestionsSP* (left)

As a further analysis, we investigate the computational advantage of relational coalescing by computing the number of paths originating from the entities mentioned in the questions both in the original KG and in the coalesced representation. Figure 3 presents the results for both *MetaQA* and *WebQuestionsSP*. The experiment shows that relation coalescing allows reducing the number of paths by up to 2 orders of magnitude on both datasets. This directly impacts both the memory requirements and the efficiency of our approach. For *MetaQA* we analyzed paths of length up to 3, whereas for *WebQuestionsSP* we consider paths of length 1 or 2, as the dataset does not include 3-hop questions.