# OpenReview forum: "SQALER: Scaling Question Answering by Decoupling Multi-Hop and Logical Reasoning"
_NeurIPS.cc/2021/Conference — NeurIPS 2021 Poster_

### Official Review · Reviewer_Rcue · 2021-07-12

**Rating:** 6
**Confidence:** 4

**Summary:**

This paper proposes an end-to-end KBQA system that can be directly used on the whole knowledge bases. This is in contrast to existing approaches that need a heuristic subgraph construction stage. More specifically, the proposed method trains a relation-following model to collect a candidate set of entities and the corresponding subgraph, which can then be processed by existing graph models. Empirically, the proposed method achieves state-of-the-art performance on MetaQA and WebQuestionsSP. Overall, this paper presents string empirical results over existing baselines on multi-hop KBQA, however, the main idea is hardly novel and has been applied to similar tasks.

**Limitations And Societal Impact:**

I don't see any negative social impact of this work.

**Main Review:**


**Originality**: The core idea of the model is not new. The idea of sequentially selecting relation links to extend the reasoning chains has been extensively explored by KB reasoning work (DeepPath) and also semantic parsing methods on Freebase (such as stagg https://www.microsoft.com/en-us/research/wp-content/uploads/2016/02/ACL15-STAGG.pdf). The proposed method uses this process as an initial stage to replace pipelines like personalized page rank.

**Quality**: The presentation is clear and the experiments are well designed. It not only shows state-of-the-art performance but also better compositional generalization.

**Clarity**:  The writing is clear and easy to follow.

**Significance**: While similar ideas have been applied to related tasks, this work shows benefits on multi-hop KBQA tasks.




**Time Spent Reviewing:**

2

---

> ### Author Response · Authors · 2021-08-10
> **Reply to Reviewer Rcue**
>
> We thank the reviewer for the overall positive feedback, for their time, and for the nice comments on the presentation and the writing of the paper.
>
> We agree that ideas similar to our work have been applied to related tasks. The papers cited by the reviewer are good examples of works that rely on multi-hop reasoning and relational following to traverse a knowledge graph. Indeed, we cited DeepPath and we will include a reference to STAGG.
>
> However, we think the main originality of our paper does not lie in the approach used to traverse the knowledge graph. We believe that the main contribution and originality of our work is showing that multi-hop reasoning can be used to generate query subgraphs that provably contain the actual answers to any existential positive first-order query. This idea, as shown in the paper, yields a number of advantages for KBQA approaches, including the possibility of scaling KBQA to the size of real-world KGs. Approaches like DeepPath apply reinforcement learning (RL) in tasks that are much more aligned to the idea of following relations to perform reasoning on a KG. Showing that a similar approach can be used to generate answer candidates for any EPFO query (possibly with multiple anchor nodes) in a scalable way is the main novelty in our paper. Also, note that relational following is not sufficient to answer EPFO queries, which is the reason why an edge-level model for logical reasoning is needed in WebQSP. During the development of our approach, we performed some preliminary experiments where the relation-level model for multi-hop reasoning and the edge-level model for logical reasoning were trained in an end-to-end way with a RL approach. This did not yield any improvement to our overall KBQA performance, but we plan to explore this idea further in future work.

---

> > ### Author Response · Authors · 2021-09-02
> > **New results**
> >
> > Dear Reviewer,
> >
> > thanks again for your work on our manuscript. We are writing to you to point to your attention that we obtained new experimental results on incomplete KGs, in light of the discussion with _Reviewer aN8i_. Unfortunately, we did not receive any reply from the reviewer, though we think that all concerns in their initial review have been majorly clarified from our rebuttal. Anyway, we believe that the new results can be of interest to all reviewers, and we would kindly ask you if you could have a look at our rebuttal and possibly update your rating based on our latest experiments.
> > We report below the following table to summarize our results on WebQSP with incomplete KGs (only 50% of the original edges), both with and without additional external knowledge in the form of a text corpus. More details are given in the reply to _Reviewer aN8i_.
> >
> > |  | 50% KB | 50% KB + Text |
> > |---|---|---|
> > | **GRAFT-Net** | 48.2 | 49.9 |
> > | **PullNet** | 50.3 | 51.9 |
> > | **SQALER - GNN** | **53.5** | **55.2** |
> >
> >
> > Thanks again for your consideration,
> >
> > Authors of the submission

---

### Official Review · Reviewer_aN8i · 2021-07-16

**Rating:** 4
**Confidence:** 4

**Summary:**

The paper focuses on the task of knowledge graph question answering. It specifically aims to address the challenge of scalability issue of the current KGQA methods. The overall idea is to first obtain a question-specific coalesced subgraph out of the knowledge graph, which “merges” entities of the same type (connected by the same relation types). Then the method locates a subset of answer candidates on the subgraph by traversing on this coalesced subgraph with some relation sequences derived from the mentioned entities in the question. Then they run a GNN model to select the final answer to the question from the answer candidates.

Generally I feel the paper writing can be refactored and improved. I suggest the authors list all the model components at the beginning of Section 2. Currently they only introduce a relation-level model and the edge-level model in Section 2, then out of nowhere, they mention they further propose an autoencoder in Section 3. They also say that “the architecture of the model includes three main components: a question encoder, a relation encoder, and a graph-guided decoder” in line 163. Where is relation-level model and edge-level model?

One key assumption (or limitation) in the paper is that the graph is complete, which renders the proposed method weaker than the baselines, e.g., all Pullnet, embedkgqa, EmQL can deal with missing edges. Although the authors mention in line 147-154 that “it may still be able to retrieve alternative sequences of relations, provided that the training data are representative enough”, can you run experiments on incomplete MovieGraph for MetaQA? I assume the training questions in MetaQA are much more than that in WebQSP and should also be very “representative” since the questions in the dataset are generated using some templates.

Another general comment is that since the paper focuses on QA in the complete graph setting and aims to obtain a relation sequence as the “parsed representation” of the question. Can you also compare your method with the series of (unsupervised) semantic parsing methods? These methods aim to parse the input questions to some logic form and then execute the logical queries on KG for answers.



**Limitations And Societal Impact:**

Please check the section above, I do not think it has potential negative societal impact.

**Main Review:**

1. I am confused by line 28-31, what exactly do you mean by “each forward pass scales at least linearly with the number of facts in the KG”? What is a forward pass? If you mean obtaining the subgraph for the question, I think in Pullnet obtaining the subgraph is not running a classifier on all the facts (edges) on the KG, but rather an iterative process, where at each step the classifier only needs to classify a small set of edges that are connected to the current retrieved subgraph. This is not linear to the number of facts on the KG and actually it is much less.
2. It seems that the paper misses one important section, which is the detailed architecture of the edge-level model.
3. Do you assume $\mathcal{V}_Q$ are given? The authors should make it clear what is given and what is not.
4. In line 83-93, the authors talk about how to obtain the coalesced graph. The term “Reachability” is conditioned on a sequence of relations as defined in the Reachability paragraph. However, in line 87, the authors only say “The nodes of $\tilde{\mathcal{G}_Q}$ are a set of nodes of $\mathcal{G}$ that are reachable from $\mathcal{V}_Q$” Then what is the relation sequence here? Do different questions have different relation sequences? How do you obtain the relation sequences? Significant amount of details are missing here and I could not find that in Appendix A.
5. The notation is confusing for the equation after line 97. As in the definition, $\phi: \tilde{\mathcal{E}_Q} \to [0,1]$ really maps an edge in $\tilde{\mathcal{G}_Q}$ to a value between 0 and 1, but in the equation after line 97, for the argument of the function $\phi$, I do not think $reach_\mathcal{G}(\mathcal{V}_Q,R_{1\to i-1})$ represents a node on the KG but rather it represents **a set of nodes** on the KG as in the definition above line 83. Can you clarify the equation?
6. Also, how long is the $R_i*$? Do you allow different lengths for $R_i*, i=1,\dots,k$?
7. I am also confused by the motivation of the knowledge seeking process. It seems that $\tilde{\mathcal{A}_Q}$ is obtained using $R_i*$, which is obtained by maximum likelihood $p(R_i*|\tilde{\mathcal{G}_Q},\mathcal{V}_Q)$. This is saying that the answer candidates in the coalesced subgraph are only conditioned on the original KG and the question entities, but not conditioned on the original question. This is rather counterintuitive since there exist many questions with drastically different answers but with the same starting entities.
8. Besides, how would you train the network $\phi$?
9. What is $\mathcal{G}(\tilde{\mathcal{A}_Q})$ and $\mathcal{V}(\tilde{\mathcal{A}_Q})$ in line 106-107? $\mathcal{G}$ and $\mathcal{V}$ are not functions.

minor:
1. I would suggest using different notations for natural language question and logical query. Currently both are denoted as Q in different fonts.
2. line 159: “We found useful” -> “We found it useful”


**Time Spent Reviewing:**

6

---

> ### Author Response · Authors · 2021-08-10
> **Reply to Reviewer aN8i**
>
> We thank the reviewer for the feedback and for the time spent on our manuscript. We believe that probably there are some misunderstandings about core parts of the paper and that the overall rating reflects more this problem rather than weaknesses of our approach. In the following, we address the main concerns raised by the reviewer and then we answer the questions in the main review by pointing to specific lines in the paper that should clarify some doubts.
>
> **Structure of the paper.** In the beginning of Section 2, we provide an overview of our approach, which consists of two components, a relation-level and an edge-level model. As noted by another reviewer, the two  models serve the purpose of performing KBQA in a coarse-to-fine way. Section 2 then details the overall approach in a way that is independent of the implementation of the two models. Section 3 does not propose any further model, as the autoencoder described in Section 3 is precisely the relation-level model. This is explained in lines 156-157. The purpose of Section 3 is to specify how the relation-level model has been implemented.
>
> **Incomplete KGs.** We agree that our approach relies on the assumption that the KG is complete. We mentioned this limitation in lines 147-149. As noted by the reviewer, PullNet and GRAFT-Net can deal with missing edges by using additional sources of information like external text documents. In principle, we can easily incorporate additional knowledge in SQALER by providing external documents as input to a GRAFT-Net edge-level model (note that our current edge-level model is the KB-only version of GRAFT-Net). _We performed this additional simple experiment, achieving 55.2% Hits@1 on WebQSP with an incomplete version of Freebase with only 50% of the original edges._ This is to be compared to a Hits@1 of 51.9 for PullNet and 49.9 for GRAFT-Net. We used the same documents as in the GRAFT-Net paper. We will mention in the limitations that external knowledge can help to deal with incomplete KGs. Other methods may be used to deal with missing edges, but reasoning on incomplete KGs is not the main focus of this paper and we may leave this to future work.
>
> **Semantic parsing.** We have already compared our method with semantic parsing approaches on CFQ. All baselines listed in the Table 2 are semantic parsing approaches, as specified in lines 233-234. As such, they require additional supervision, as they need a parsed representation of the question, whereas our approach only requires the answer nodes. To the best of our knowledge, the top-performing approach for KBQA with semantic parsing on WebQSP is the recent CBR-KBQA [1]. _This approach achieves a KBQA performance of 72.8 on WebQSP, which is below our performance of 76.1._
>
> In the following, we reply in more detail to the specific points raised in the main review. We will make an effort to clarify these points and improve the structure of the paper in the camera ready version.
>
> 1. Thanks for your question. In line 28-31, by “forward pass” we mean applying a single neural network on a KG for KBQA. Hence, we refer to a neural network that classifies the nodes of the graph (the full KG or a subgraph) as either being answers to the input question or not. In PullNet, this step still scales linearly with the number of edges in the graph, as it relies on GRAFT-Net (i.e., a GNN). Note that, in lines 27-28, we cite both PullNet and GRAFT-Net and we specify that, due to such linear complexity, PullNet needs a preprocessing step to build a small query-dependent subgraph using an “iterative procedure based on learned classifiers”. This is exactly the iterative process mentioned by the reviewer. We will rephrase lines 28-31 to make clear what we mean and avoid any further confusion.
>
> 2. We did not include a detailed architecture of the edge-level model because we use well-known approaches for the refining step (a GNN and a Key-Value Memory Network). The best performing model (SQALER – GNN) employs a GNN with the same architecture as GRAFT-Net, as explained in Section 2.2. We may include a detailed architecture in the Appendix of the camera ready version.
>
> 3.  Thanks for the useful comment. Yes, we assume that the set of anchor nodes is given in our experiments, as all the baselines rely on the same assumption. We will specify this more explicitly in the beginning of Section 2.
>
> 4. The nodes of the coalesced graph are sets of nodes of the original graph that are reachable from $\mathcal{V}_Q$ by following all sequences of relations originating from $\mathcal{V}_Q$, as specified formally in Appendix A.
> Note that, in Appendix A, the nodes of $\tilde{\mathcal{G}}_Q$ are defined as $\tilde{\mathcal{V}}_Q = \\{reach(\mathcal{V}_Q, R) \mid R \in \mathcal{R}^* \\}$ so the set includes all nodes R-reachable from $\mathcal{V}_Q$ by all sequences $R \in \mathcal{R}^*$ (* is the Kleene star).
> #
> This is also visually shown in Figure 1. However, as explained both in Appendix A, Section 2.2 and Section 2.3, we do not actually build the full coalesced graph, but we only collect all relation sequences of length up to $\tau_{\textit{max}}$. Appendix B further details that we set $\tau_{\textit{max}} = 2$ for MetaQA 2 and WebQSP and we set  $\tau_{\textit{max}} = 3$  for MetaQA 3.
>
> 5. The relation-level model $\phi$ maps an edge of the coalesced KG $\tilde{\mathcal{G}}_Q$ to a value between 0 and 1.
> Also, the _reach_ function returns a set of nodes of the original KG, yet there is no contradiction in the equation.
> Indeed, remember that, as specified in line 87 and shown in Figure 1, the nodes of $\tilde{\mathcal{G}}_Q$ are precisely sets of nodes of the original KG. The set of nodes of the coalesced graph is also defined formally in Appendix A.
>
> 6. We allow different lengths for the relation sequences $R_i^*$. As explained in Section 3, the relation sequences leading from the entities mentioned in the question to the answers are predicted by the graph-guided decoder one relation at a time. Hence, the length of $R_i^*$ is bounded by the maximum number of decoding time steps, but the approach can predict shorter sequences, as better explained in Appendix B (lines 36-37) and detailed in Algorithm 1.
>
> 7. The relation-level model $\phi$, which is used to compute the likelihood of the different relation sequences, is the autoencoder described in Section 3 and depicted in Figure 2. Hence, it is conditioned on the question. We will make this explicit in the equation after line 97.
>
> 8. We train the network $\phi$ as explained in lines 156 to 163. We rely on weak supervision, assuming that a sequence of relations is correct if it reaches the smallest superset of the actual answers. As mentioned in the paper, Appendix D provides more details about training strategies.
>
> 9. In lines 106-107, we use the notation $\mathcal{G}(\tilde{\mathcal{A}}_Q)$ to identify the subgraph that the edge-level model is applied on. Similarly, $\mathcal{V}(\tilde{\mathcal{A}}_Q)$ denotes the nodes of this graph. We used this as a compact notation to remind the reader that this subgraph is constructed based on the candidates $\tilde{\mathcal{A}}_Q$ predicted by the relation-level model. We will consider changing the notation in the camera-ready version of the paper.
>
> [1] Rajarshi Das, Manzil Zaheer, Dung Thai, Ameya Godbole, Ethan Perez, Jay-Yoon Lee, Lizhen Tan, Lazaros Polymenakos, Andrew McCallum. Case-based Reasoning for Natural Language Queries over Knowledge Bases. https://arxiv.org/abs/2104.08762

---

> > ### Comment · Reviewer_aN8i · 2021-08-22
> > **Thank you**
> >
> > Thank you for the response and all the new experimental results. I have some additional questions listed below.
> >
> > > Incomplete KGs
> >
> > Thank you for the additional experiments. But I am not fully convinced of your response that PullNet can only deal with missing edges by using external document. The iterative retrieval process can also provide signals because although the answer node may not be directly reachable due to the missing edge, it can still be included in the subgraph by performing several rounds of retrieval. They also demonstrate the performance of the model on incomplete KGs without the external document. I wonder if the author can justify how/whether the proposed model can be improved to handle these missing facts and show some results of the model on incomplete KG without using external knowledge if possible.
> >
> > > Hence, we refer to a neural network that classifies the nodes of the graph (the full KG or a subgraph) as either being answers to the input question or not. In PullNet, this step still scales linearly with the number of edges in the graph, as it relies on GRAFT-Net (i.e., a GNN).
> >
> > Sorry I am not sure whether this statement is accurate since the number of edges in the subgraph of each question is much smaller than the whole graph. For example, the number of nodes of the retrieved graph on average for WebQuestionsSP on Freebase is 1876.9 (Table 4 in the PullNet paper), however, the Freebase has more than 24.9 million entities and 164.6 million facts. I am pretty sure the number of edges in the subgraph does not scale linearly with the number of facts in the KG (line 29 in the submission).
> >
> > Besides, can you also clarify your statement that "a method that scales linearly with the number of relation types in the KG"? I think this really depends on the $\tau_{max}$. If you consider a three-hop path, then the size of the coalesced graph will be $\mathcal{O}(R^3)$ right?
> >
> > > We did not include a detailed architecture of the edge-level model because we use well-known approaches for the refining step (a GNN and a Key-Value Memory Network). The best performing model (SQALER – GNN) employs a GNN with the same architecture as GRAFT-Net, as explained in Section 2.2. We may include a detailed architecture in the Appendix of the camera ready version.
> >
> > Thank you, please make sure the paper is self-contained.
> >
> > > The relation-level model , which is used to compute the likelihood of the different relation sequences, is the autoencoder described in Section 3 and depicted in Figure 2. Hence, it is conditioned on the question. We will make this explicit in the equation after line 97.
> >
> > Sorry from the equation between line 97 and 98, I do not see how $p(R_i*|\tilde{\mathcal{G}_Q},\mathcal{V}_Q)$ is conditioned on the question. For example both questions 'where was Obama born?' and 'what award has Obama won?' have the exact same entity mention $V_Q$ being Obama, and will also have the same coalesced graph $\tilde{G_Q}$. Also what $\phi$ does is to map an edge to [0,1] (line 95). I do not see how the knowledge seeking process is different for the two questions.

---

> > > ### Author Response · Authors · 2021-08-24
> > > **Thank you**
> > >
> > > Thanks for your kind reply and the additional feedback. We would really like to express our gratitude for engaging with us in conversation to clear up the issues and improve the paper. In the following, we answer the questions in your comment.
> > >
> > >
> > > > Incomplete KGs.
> > >
> > > We agree that the subgraphs built by PullNet can still contain the answer node even if some edges in the KG are missing. This can only happen if there exists a path between an anchor node mentioned in the question and an answer node. Hence, any node that can be found by PullNet’s iterative retrieval process is reachable from the set of anchor nodes and, in principle, can also be retrieved by SQALER’s relation-level model. We performed an additional experiment on WebQSP, using only 50% of the original edges and no external text document. We achieved a performance of 53.5% Hits@1, which is to be compared to 50.3% for PullNet and 48.2% for GRAFT-Net. Hence, SQALER outperforms both PullNet and GRAFT-Net even on incomplete KGs. This is not surprising, as GRAFT-Net relies on a simple heuristic process to construct the subgraphs and PullNet (like SQALER) is constrained to follow the structure of the (incomplete) graph, as discussed above. A possible way to further enhance SQALER’s performance on incomplete KGs could be to include node-level features in the representation of the coalesced graph, using pre-trained KG embeddings combined with an effective set representation (like the sparse-dense encoding used by EmQL). We did not perform this experiment as it is out of the scope of our paper, but we will consider investigating this research path in future work. The following table summarizes our new results on incomplete KGs.
> > >
> > > |  | 50% KB | 50% KB + Text |
> > > |---|---|---|
> > > | **GRAFT-Net** | 48.2 | 49.9 |
> > > | **PullNet** | 50.3 | 51.9 |
> > > | **SQALER - GNN** | **53.5** | **55.2** |
> > >
> > >
> > >
> > > > Sorry I am not sure whether this statement is accurate since the number of edges in the subgraph of each question is much smaller than the whole graph. [...] I am pretty sure the number of edges in the subgraph does not scale linearly with the number of facts in the KG (line 29 in the submission).
> > >
> > > Line 29 does not state that the number of edges in the subgraph scales linearly with the number of edges in the KG. It states that applying a single neural network on a graph for KBQA has a computational complexity that is linear in the number of edges of the graph that the network is applied on. Whether the graph is the full KG or a subgraph of it, the complexity of this process is still linear in the size of that graph. In PullNet, the final classification step is linear in the size of the subgraph built by the iterative retrieval process, because it relies on a GNN (GRAFT-Net) that is applied on that subgraph. Line 29 is just meant to explain why methods like PullNet and GRAFT-Net (both cited in lines 27-28) need to construct a small query-dependent subgraph. We will rephrase that paragraph to avoid any confusion.
> > >
> > > > Besides, can you also clarify your statement that "a method that scales linearly with the number of relation types in the KG"? I think this really depends on the $\tau_{\textit{max}}$. If you consider a three-hop path, then the size of the coalesced graph will be $\mathcal{O}(|\mathcal{R}|^3)$ right?
> > >
> > > The coalesced graph is explored using an algorithm based on the beam search and it does not need to be loaded completely, neither at training nor at inference time. Actually, we do not even have to fully construct it, as mentioned in lines 91-92 and better explained in Appendix B. Since the knowledge seeking procedure is not an exhaustive search, for a three-hop path the complexity is not $\mathcal{O}(|\mathcal{R}|^3)$, but it is rather $\mathcal{O}(|\mathcal{R}| \cdot \beta \cdot 3) = \mathcal{O}(|\mathcal{R}| \cdot \beta)$, where $\beta$ is the beam width.
> > >
> > > > Sorry from the equation between line 97 and 98, I do not see how $\mathsf{P}(R \mid \tilde{\mathcal{G}}_Q, \mathcal{V}_Q)$ is conditioned on the question. For example both questions 'where was Obama born?' and 'what award has Obama won?' have the exact same entity mention $\mathcal{V}_Q$ being Obama, and will also have the same coalesced graph $\tilde{\mathcal{G}}_Q$. Also what $\phi$ does is to map an edge to [0,1] (line 95). I do not see how the knowledge seeking process is different for the two questions.
> > >
> > > The knowledge seeking procedure is different for the two examples because the model $\phi$ is conditioned on the question. In our implementation, $\phi$ denotes the output of the autoencoder depicted in Figure 3, which uses the question encoder to take different questions into account in the knowledge seeking procedure. As mentioned in our previous reply, we will make this explicit in the equation after line 97, by rewriting it as
> > >
> > > $$
> > > \mathsf{P}(R \mid Q, \tilde{\mathcal{G}}_Q, \mathcal{V}_Q) \propto \prod \phi(\dots \mid Q),
> > > $$
> > >
> > > so that it should be clear that both sides are conditioned on the question. Even if the coalesced graph and the set of entity mentions are the same, the model $\phi$ assigns different weights to the edges based on the input question.

---

> > > > ### Author Response · Authors · 2021-08-28
> > > > **Thank you**
> > > >
> > > > Dear reviewer,
> > > >
> > > > Your original rating was supported by concerns that have been majorly clarified after our discussion.
> > > >
> > > >  Our additional experiments also address the main limitation of our original manuscript, showing that our approach can handle missing edges better than relevant baselines.
> > > >
> > > > We thus ask you if you would consider increasing your rating appropriately.
> > > >
> > > > We commit to update the camera-ready version to clarify these matters for the benefit of future readers.
> > > >
> > > > We thank you for your reconsideration and willingness to engage in productive dialogue.

---

> > > > > ### Author Response · Authors · 2021-09-01
> > > > > **Reply to Reviewer aN8i**
> > > > >
> > > > > Dear Reviewer,
> > > > >
> > > > > we would like to kindly ask you if there are any remaining concerns that we did not address in our previous reply. If so, we are open to clarify any issue and engage in productive dialogue.
> > > > >
> > > > > Many thanks,
> > > > >
> > > > > Authors of the submission

---

### Official Review · Reviewer_SVCA · 2021-07-16

**Rating:** 7
**Confidence:** 3

**Summary:**

This paper proposed a KBQA framework that scales linearly with the number of relation types in a knowledge graph. The framework contains three main components in general. They first adopt relational coalescing to combine the nodes which share the same relation to reduce the number of edges in a graph. Then, they utilized a relation-level model to extract a subgraph that contains a set of answer candidates. Finally, the answer candidates are refined by an edge-level model. Their experiments on knowledge-based question answering show that the proposed framework achieves a new state-of-the-art on the challenging WebQuestionsSP, and is orders of magnitude more scalable than competitive approaches.

**Limitations And Societal Impact:**

The authors sufficiently addressed the limitations of their work and I don't think there are potential negative societal impacts.

**Main Review:**

Personally, I liked the idea of doing KBQA in a coarse-to-fine way. The proposed relational coalescing is simple and intuitive, yet very effective. The experimental results on the common datasets are quite strong and also show that the proposed method generalizes compositionally. These are all good merits that could spur further researches of KBQA in this coarse-to-fine fashion. Also, the paper is well-presented and I enjoyed the reading.

Here are several questions I have:
1. What is the average length of the relational chains for WebQSP?
2. Could the proposed framework generalize to the question that contains several entities?
3. Are there some typical cases where the relational-level model fails to retrieve the correct chain?

**Time Spent Reviewing:**

4-5

---

> ### Author Response · Authors · 2021-08-10
> **Reply to Reviewer SVCA**
>
> We thank the reviewer for the positive comments and the thorough review. We answer below the main questions asked by the reviewer.
>
> 1. The average length of relational chains for WebQSP is 1.38. Most questions in WebQSP are 1-hop questions. The 2-hop questions in the dataset always involve relational chains containing CVT (compound value type) nodes, which are used in Freebase to express n-ary relations. Some examples are shown in Appendix E.4.
> 2. Yes, the proposed framework is designed to handle questions containing one or more anchor entities, as this is required for some questions in WebQSP. Figure 1 in Appendix C.2 shows how the framework can handle this kind of questions.
> 3. We performed several error analyses when developing the model, but we could not find typical cases where the relation-level model fails to retrieve the correct chain. The model reaches a similar performance both on 1-hop and 2-hop questions and the same holds for questions with only one or with multiple anchor entities.

---

> > ### Author Response · Authors · 2021-09-02
> > **New results**
> >
> > Dear Reviewer,
> >
> > thanks again for your work on our manuscript. We are writing to you to point to your attention that we obtained new experimental results on incomplete KGs, in light of the discussion with _Reviewer aN8i_. Unfortunately, we did not receive any reply from the reviewer, though we think that all concerns in their initial review have been majorly clarified from our rebuttal. Anyway, we believe that the new results can be of interest to all reviewers, and we would kindly ask you if you could have a look at our rebuttal and possibly update your rating based on our latest experiments.
> > We report below the following table to summarize our results on WebQSP with incomplete KGs (only 50% of the original edges), both with and without additional external knowledge in the form of a text corpus. More details are given in the reply to _Reviewer aN8i_.
> >
> > |  | 50% KB | 50% KB + Text |
> > |---|---|---|
> > | **GRAFT-Net** | 48.2 | 49.9 |
> > | **PullNet** | 50.3 | 51.9 |
> > | **SQALER - GNN** | **53.5** | **55.2** |
> >
> >
> > Thanks again for your consideration,
> >
> > Authors of the submission

---

### Decision · Program_Chairs · 2021-09-27

**Decision:**

Accept (Poster)

**Comment:**

This work proposes a substantially more scalable approach to KBQA that scales with the number of relation types (rather than the number of edges) in a KG, and operates in a coarse-to-fine manner.  The overall idea of working first at relation types, and later refining to individual edges, is interesting, and the results in the setting considered are strong.

The detailed discussion during the reviewing + author response period clarified some aspects (e.g., that the main contribution is not traversing a KG for multi-hop reasoning but identifying in a coarse-to-fine and scalable way to identify subgraphs that contain the answer), and also included additional positive results such as improved performance even on incomplete KGs, which is what the PullNet and GraftNet baselines were originally designed for.

The paper is close to the fence, with several strengths and some weaknesses:

**Strengths**: Strong empirical performance, intuitive overall idea, and authors' active effort in addressing (and, I assume, incorporating in the revised draft) detailed feedback from the reviewers.

**Weaknesses**: Limited novelty (as many pieces of the overall idea exist), room for clarity of presentation of both technical details (some of which are in the appendix) and broad statements like existing methods scaling linearly in the size of the entire KG (whereas they scale linearly in the size of the *retrieved* subgraph, which is often much smaller for 2-4 hop questions).